# ROUTING MANIFOLD ALIGNMENT IMPROVES GENERALIZATION OF MIXTURE-OF-EXPERTS LLMS

**Zhongyang Li**
Johns Hopkins University
Baltimore, MD, USA
`zli300@jh.edu`

**Ziyue Li**
University of Maryland
College Park, MD, USA
`litzy619@umd.edu`

**Tianyi Zhou**
MBZUAI
Abu Dhabi, UAE
`tianyi.zhou@mbzuai.ac.ae`

## ABSTRACT

Sparse Mixture-of-Experts (MoE) have been widely adopted in recent large language models since it can efficiently scale up the model capability without increasing the inference cost. However, evaluations on broad downstream tasks reveal a consistent suboptimality of the routers in existing MoE LLMs, which results in a severe performance gap (e.g., 10-20% in accuracy) to the optimal routing. In this paper, we show that aligning the manifold of routing weights with that of task embedding via post-training can effectively reduce the gap and improve MoE LLMs' generalization performance. Our method, "**Ro**uting **M**anifold **A**lignment (RoMA)", introduces an additional manifold regularization term in the post-training objective and only requires lightweight finetuning of routers (with other parameters frozen). Specifically, the regularization encourages the routing weights of each sample to be close to those of its successful neighbors (whose routing weights lead to correct answers) in a task embedding space. Consequently, samples targeting similar tasks will share similar expert choices across layers. Building such bindings between tasks and experts over different samples is essential to achieve better generalization. Moreover, RoMA demonstrates the advantage of unifying the task understanding (by embedding models) with solution generation (by MoE LLMs). In experiments, we finetune routers in three recent MoE LLMs using RoMA. Evaluations on diverse benchmarks and extensive comparisons with baselines show the substantial improvement brought by RoMA. Our code can be accessed here.

## 1 INTRODUCTION

Sparse Mixture-of-Experts (MoE) have emerged as a cornerstone architecture in scaling large language models (LLMs), enabling significant capacity increases without proportional computational overhead during inference (Fedus et al., 2022; Lepikhin et al., 2020). At the core of this mechanism lies the router, which assigns input tokens to a small subset of experts through routing weights in each layer. Despite the small portion of router parameters in MoE LLMs (e.g., 0.03% in a 7B model), they are the key to the success of expert usage in MoE (Shazeer et al., 2017). However, evaluations across broad downstream tasks reveal that routers in existing MoE LLMs cause major failures. As shown in Table 1, their suboptimal routing weights lead to a performance gap of 10-20% in accuracy when compared to the optimal routing weights (oracle). This gap underscores a major untapped bottleneck in MoE LLMs, suggesting that improving routing is critical to boosting MoE LLMs' generalization performance on downstream tasks.

Recent findings on "grokking" in LLM pretraining reveal that routing pathways naturally evolve to become more structured and consistent across layers during the transition from memorization to generalization (Li et al., 2025c). Building on this insight, our analysis further investigates the reasons behind the performance gap and the poor generalization capabilities of pretrained routers. As illustrated in Figures 3(a) and (b), pretrained routers assign semantically similar samples in the task embedding space to distinct experts with dramatically different routing weights. Such misalignment between the task embedding manifold and routing weight manifold hinders effective knowledge sharing across tasks and underutilize the collective expertise of the experts. This misalignment between the targeted tasks and the assigned experts undermines the generalization of MoE and its

core principle, which is to leverage specialized experts, share skills, and transfer knowledge for related inputs.

A natural solution is to finetune the routers. Existing approaches such as Dense BP (Panda et al., 2025) developed more effective pretraining objectives for routers but do not address the manifold misalignment between the targeted tasks and the routing weights across samples.

This limitation motivates our exploration of incorporating manifold alignment into the fine-tuning objective. Specifically, our manifold alignment aims to enforce the consistency between task understanding (encoded by an embedding model) and task solving in an MoE LLM (encoded by the routing weights). As illustrated in Figure 2, for each training sample, in addition to minimizing its loss defined on the output, we encourage its intermediate layers' routing weights to move to those of its "*successful neighbors*" (samples with correct MoE predictions) in the task embedding space. These neighbors are weighted by their similarity to the sample. This training objective can be formulated as manifold regularization (Belkin et al., 2006), a well-established technique in machine learning that aims to preserve the local neighborhood structure of high-dimensional inputs on the manifold of low-dimensional representations or outputs. Unlike its original setting, we apply such a regularization to the routing weights across MoE layers rather than the final outputs, and establish coherent bindings between the expert choices (weights) and the task embedding instead of the raw inputs.

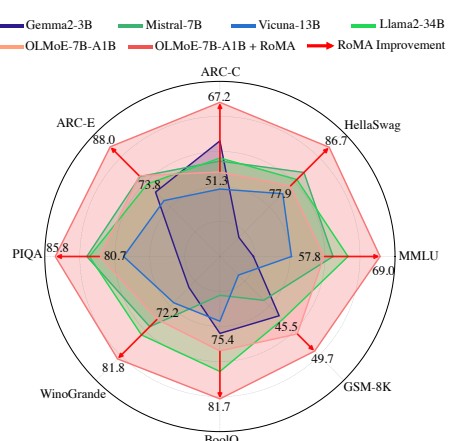

Figure 1: RoMA on OLMoE-7B-A1B vs. 7-34B dense LLMs across eight benchmarks. RoMA leads to 7-15% accuracy improvement, consistently outperforming all models over eight benchmarks, demonstrating the effectiveness of post-training by RoMA.

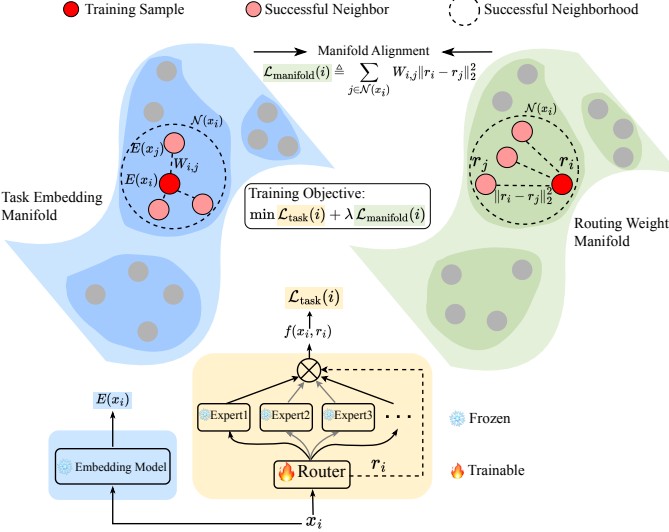

Figure 2: Overview of RoMA. RoMA finetunes routers in MoE LLM (bottom, yellow) with a training objective defined on each sample $(x_i, y_i)$, which is composed of (1) the task loss $\mathcal{L}_{\text{task}}(i)$ defined on the model output $f(x_i, r_i)$; and (2) the manifold alignment regularization $\mathcal{L}_{\text{manifold}}(i)$, which aligns the manifolds of routing weights (right, green) and the task embedding (left, blue). It improves MoE's generalization by unifying solution generation in MoE with task understanding.

To this end, we propose "**Ro**uting **M**anifold **A**lignment (RoMA)", a router post-training method that aligns the manifold of routing weights with task embeddings through lightweight fine-tuning of a few routers in MoE LLMs. RoMA introduces a manifold regularization term to the training objective that encourages routing weights of each sample to approximate those of its successful neighbors with similar task embedding, thereby promoting consistent expert selection for semantically related inputs. Extensive experiments on three recent MoE LLMs (OLMoE, DeepSeekMoE, Qwen3-MoE) demonstrate that RoMA brings substantial improvements (7-15% in accuracy) across diverse benchmarks and outperforms SOTA routing methods, as shown in Figure 1, by merely finetuning 0.0095% parameters of the base model, without affecting inference cost.

Notably, RoMA-finetuned MoE LLMs with only 1-3B active parameters achieve competitive or superior performance over much larger dense models with 34B parameters. We conduct comprehensive ablation studies that further investigate the effects of key designs in RoMA, including layer selection,

neighborhood configuration, and regularization strategies, validating the effectiveness of RoMA in bridging the performance gap between pretrained routers and optimal routing for MoE LLMs.

## 2 RELATED WORK

**MoE LLMs** Mixture of Experts (MoE) architectures have been extensively incorporated into large language models (LLMs) to enhance computational efficiency and task-specific specialization (Shazeer et al., 2017). Recent work such as OLMoE (Muennighoff et al., 2024) and DeepSeek-MoE (Dai et al., 2024a) demonstrate the effectiveness of sparse MoE layers in reducing active parameters while maintaining model capacity. These MoE models fundamentally rely on routers to determine expert selection, typically employing token-choice routing that selectively activates subsets of experts for each input token (Fedus et al., 2022; Lepikhin et al., 2020). Beyond training MoE models from scratch, MoEfication (Zhang et al., 2022) proposes converting pretrained dense models into MoE architectures by splitting feed-forward network parameters into functional partitions as experts. However, the quality of these routing decisions remains a critical bottleneck. Our study shows current routers often produce suboptimal routing weights that fail to fully leverage expert specialization, resulting in load imbalance and expert underutilization.

**Manifold Regularization of LLMs** Recent work reveals that LLM embeddings exhibit stratified manifold structures with varying dimensions across semantic domains (Li & Sarwate, 2025; Robinson et al., 2025). Furthermore, recent studies show that the routing weights of MoE LLMs inherently capture rich semantic representations and can serve as effective off-the-shelf embedding models (Li & Zhou, 2025). While traditional manifold regularization assumes smooth global structures (Belkin et al., 2006), LLMs require more sophisticated approaches. Methods like I-STAR (Rudman & Eickhoff, 2023) control isotropy in embedding spaces, while CROW (Min et al., 2024) enforces consistency across layers. However, these techniques do not explicitly leverage manifold structures to improve MoE routing. The geometric insights from stratified manifolds suggest that different experts naturally align with different embedding strata, yet current routing mechanisms fail to exploit this alignment. This gap motivates our routing manifold alignment approach, which guides routing decisions based on the data's inherent geometric structure.

**Routing Optimization** in MoE architectures has emerged as a critical component for achieving efficient expert utilization and balanced computation. Routing optimization methods have evolved from simple load balancing (Fedus et al., 2022; Lepikhin et al., 2020) to sophisticated strategies including differentiable top-k selection (Zhou et al., 2022) and test-time optimization such as C3PO (Li et al., 2025a) and R2-T2 (Li et al., 2025b) that dynamically re-weights expert pathways. However, these approaches optimize routing without considering the embedding space's geometric structure. Moreover, C3PO introduce additional computational overhead for task embedding and nearest neighbor search, requiring 6-7x the cost of standard inference by the base model.

## 3 TASK-EXPERT ROUTING MANIFOLD MISALIGNMENT

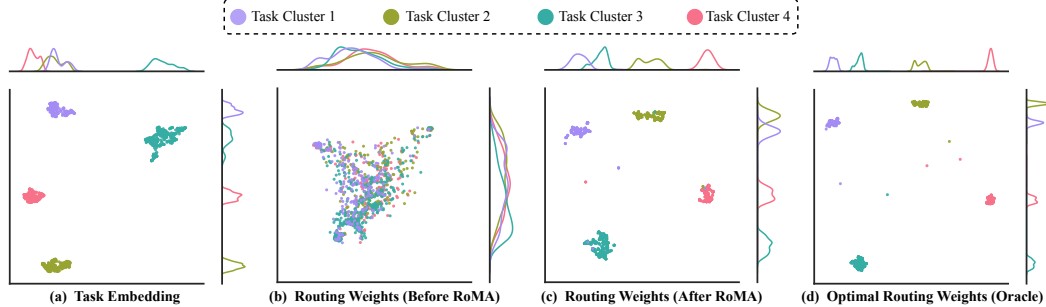

Figure 3: UMAP visualization of task embedding and routing weights manifolds for samples in ARC-C. **(a)** Their task embedding shows cluster structures. **(b)** Routing weights by pretrained MoE are scattered and misaligned with the task embedding clusters. **(c)** RoMA aligns routing weights with the task embedding manifold's cluster structure. **(d)** RoMA also achieves a similar manifold structure as that of the optimal routing weights (oracle), which explains the improvement in generalization.

MoE LLMs employ routers to assign input tokens to a small subset of experts through routing weights in each layer. For a sample $(x_i, y_i)$, let $h_i^{(\ell)}$ denote the hidden representation at layer $\ell$. The router at each layer $\ell$ produces a routing weight vector:

$$r_i^{(\ell)} = R(h_i^{(\ell)}; \theta_{\text{router}}^{(\ell)}) \in \mathbb{R}^K \tag{1}$$

where $K$ is the number of experts, $R(\cdot; \cdot)$ denotes the router function, and $\theta_{\text{router}}^{(\ell)}$ represents the router parameters at layer $\ell$. The routing weight matrix $r_i$ is computed as the concatenation of routing weights across $L$ MoE layers:

$$r_i = [r_i^{(1)}; r_i^{(2)}; \ldots; r_i^{(L)}] \in \mathbb{R}^{L \times K} \tag{2}$$

Throughout this paper, $r_i$ denotes the *routing weights* (the softmax output used to aggregate expert outputs for sample $x_i$), while $\theta_{\text{router}}$ denotes the *router parameters* (the learnable weights in the router network). Our method optimizes $\theta_{\text{router}}$ to align the manifold of $r_i$ with task embeddings.

As shown in Table 1, evaluations across broad downstream tasks reveal that routers in existing MoE LLMs produce suboptimal routing weights $r_i$, which lead to a performance gap of 10-20% in accuracy when compared to the optimal routing weights (oracle) $r_i^*$:

$$r_i^* \triangleq \arg\min_r \mathcal{L}_{\text{CE}}(f(x_i, r), y_i), \tag{3}$$

where $f(\cdot, \cdot)$ represents the MoE model that takes input $x_i$ and routing weight vector $r$ to produce output, $y_i$ is the ground truth label for input $x_i$, and $\mathcal{L}_{\text{CE}}$ is the cross-entropy loss. We obtain $r_i^*$ by initializing $r$ with the pretrained routing weights and performing gradient descent with access to the ground truth label until convergence. This oracle serves as an empirical upper bound to quantify the untapped potential of existing routers.

To investigate the root causes behind the observed performance gap in MoE LLMs, we conduct a comprehensive analysis of the relationship between task embeddings and routing weights in Figure 3.

The comparison between task embeddings (Figure 3(a)) and pretrained routing weights (Figure 3(b)) reveals a severe misalignment. While the task embedding space presents clear cluster structures where semantically similar samples are grouped together, the pretrained routing weights show no corresponding clustering patterns. Instead, samples from the same semantic cluster are scattered across the routing weights space. This manifold misalignment indicates that the pretrained routers fail to capture the underlying task structure, leading to inconsistent expert selection for semantically related inputs. To further substantiate this visual intuition, we provide quantitative alignment metrics (e.g., CKA similarity, Trustworthiness) in Appendix A.3, which consistently confirm the misalignment in baselines and the improvement brought by RoMA.

In contrast, the oracle routing weights (Figure 3(d)) demonstrate clear cluster structure to the task embedding structure, with samples from the same semantic group receiving similar routing patterns. This alignment between task understanding and expert assignment is precisely what enables the oracle to achieve superior performance, highlighting that the task-expert routing manifold misalignment is the key bottleneck limiting router generalization in MoE LLMs.

## 4 ROUTING MANIFOLD REGULARIZATION (RoMA)

To address this limitation, we propose "**Ro**uting **M**anifold **A**lignment (RoMA)", a post-training method that aligns the manifold of routing weights with task embeddings through lightweight router fine-tuning. Our key insight is that samples with similar task embeddings should share similar routing patterns to leverage specialized expertise effectively. To achieve this, we introduce a manifold regularization term that encourages alignment between the routing weight manifold and the task embedding manifold. Given a training set $\mathcal{D} = \{(x_i, y_i)\}_{i=1}^n$ and their associated routing weights $\{r_i\}_{i=1}^n$ (where $r_i$ denotes the concatenated routing weights across multiple layers), our goal is to optimize the routers such that samples with similar task embeddings share similar routing patterns.

## 4.1 SUCCESSFUL NEIGHBORHOOD TO IMITATE

We first identify the subset of training samples where the MoE produces correct predictions:

$$\mathcal{S} = \{j \in [n] : f(x_j, r_j) = y_j\} \tag{4}$$

This filtering ensures that our finetuning only imitates from routing patterns for samples in $\mathcal{S}$ that lead to successful outputs, preventing the propagation of suboptimal routing strategies.

Given the set of successful samples $\mathcal{S}$, we construct a neighborhood $\mathcal{N}(x_i)$ for each sample $x_i$ based on the task similarity in an embedding space. Let $E(\cdot)$ denote a pre-trained embedding model that maps input task descriptions/instructions to a semantic representation space. The neighborhood of $x_i$ can be defined via $k$-Nearest Neighbors or $\epsilon$-ball:

$$k\text{-NN: } \mathcal{N}(x_i) = \arg \max_{A \subseteq \mathcal{S}, |A| \leq k} \sum_{j \in A} \text{sim}(E(x_i), E(x_j)) \tag{5}$$

$$\epsilon\text{-ball: } \mathcal{N}(x_i) = \{j \in \mathcal{S} : \text{sim}(E(x_i), E(x_j)) \geq \epsilon\} \tag{6}$$

where $\text{sim}(\cdot, \cdot)$ is a similarity metric, for example, the Gaussian similarity is defined as

$$\text{sim}(E(x_i), E(x_j)) = \exp\left(-\frac{\|E(x_i) - E(x_j)\|_2^2}{2\sigma^2}\right). \tag{7}$$

## 4.2 TRAINING OBJECTIVE WITH MANIFOLD REGULARIZATION

Having identified the successful neighborhood for each sample, our next step is to incorporate this structure into the training objective to align routing behaviors with the task embedding geometry. The key idea is that semantically similar samples should not only cluster in the embedding space but also share consistent routing patterns. To achieve this, we introduce a manifold regularization term that explicitly aligns the routing weights manifold with the task embedding manifold by encouraging samples to follow the routing patterns of their successful neighbors, weighted by their semantic similarity.

The (normalized) adjacency $W_{i,j}$ between sample $x_i$ and $x_j$ is defined as

$$W_{i,j} \triangleq \frac{\text{sim}(E(x_i), E(x_j))}{\sum_{j \in \mathcal{N}(x_i)} \text{sim}(E(x_i), E(x_j))}, \quad \forall j \in \mathcal{N}(x_i), \tag{8}$$

where higher weights indicate stronger semantic similarity in the task embedding space. Given $W_{i,j}$, the manifold regularization applied to the routing weight $r_i$ of sample $x_i$ is defined as

$$\mathcal{L}_{\text{manifold}}(i) \triangleq \sum_{j \in \mathcal{N}(x_i)} W_{i,j} \|r_i - r_j\|_2^2. \tag{9}$$

By penalizing routing discrepancies $\|r_i - r_j\|_2$ between semantically similar samples with large $W_{i,j}$, $\mathcal{L}_{\text{manifold}}(i)$ enforces the routing weights manifold to be aligned with the task embedding manifold. Moreover, it moves each sample's routing weights to those of its "successful neighbors" in the task embedding space.

As a consequence, the manifold regularization consolidates the bindings between tasks and their expert choices, and thus improves the generalization.

To ensure that aligned routing patterns also lead to correct predictions, the training objective in RoMA applies the manifold regularization to the cross-entropy loss $\mathcal{L}_{\text{CE}}$ defined on the outputs.

$$\mathcal{L}_{\text{task}}(i) = \mathcal{L}_{\text{CE}}(f(x_i, r_i), y_i). \tag{10}$$

With a regularization coefficient $\lambda \geq 0$, the final objective on sample $x_i$ is

$$\mathcal{L}_{\text{RoMA}}(i) = \mathcal{L}_{\text{task}}(i) + \lambda \cdot \mathcal{L}_{\text{manifold}}(i) \tag{11}$$

During training, we only update router parameters while keeping all expert parameters frozen. The gradient update is performed via backpropagation on $\mathcal{L}_{\text{RoMA}}$ with respect to router parameters:

$$\theta_{\text{router}}^{(t+1)} = \theta_{\text{router}}^{(t)} - \eta \nabla_{\theta_{\text{router}}} \mathcal{L}_{\text{RoMA}}, \tag{12}$$

where $\theta_{\text{router}}$ represents the parameters of routers and $\eta$ is the learning rate. While router parameters represent only a small fraction of the total model parameters (0.0095%), we empirically find that only finetuning routers in the last five layers achieves superior performance while significantly saves the training cost, as demonstrated in Figure 6.

## 5 EXPERIMENTS

### 5.1 EXPERIMENTAL SETTINGS

**Models** We evaluate three recent MoE LLMs: OLMoE-7B-A1B, DeepSeekMoE-16B-A3B, and Qwen3-30B-A3B. OLMoE features a 16-layer transformer with 64 experts per layer, activating 8 per token, totaling 6.9B parameters with 1.3B active per token. DeepSeekMoE uses a 28-layer transformer with 2 shared and 64 routed experts per layer, activating all shared plus 6 routed experts per token, totaling 16.4B parameters with 2.8B active per forward pass. Qwen3-30B-A3B employs a 48-layer transformer with 128 experts per layer, activating 8 per token, totaling 30.5B parameters with 3.3B active per token. These models exemplify distinct MoE designs and scales, enabling a comprehensive evaluation of routing dynamics and generalization behavior.

**Baselines** We evaluate RoMA against both different adaptation methods (See Table 1) and other models (See Table 2) across eight benchmarks. For adaptation methods, we compare with: (1) In-Context Learning (ICL) (Brown et al., 2020) with embedding-based retrieval for few-shot demonstrations; (2) Router Tuning that directly updates the routers; (3) Oracle Tuning that fine-tunes routers with access to optimal routing weights (oracle); (4) Prefix Tuning (Li & Liang, 2021) and Soft Prompt Tuning (Lester et al., 2021) that introduce lightweight trainable parameters while keeping the base model frozen; (5) Dense Backpropagation (Dense BP) (Panda et al., 2025) that enables gradient flow through the full model while updating few parameters; (6) C3PO (Li et al., 2025a), a state-of-the-art test-time routing weights optimization method. For model comparison, we evaluate against models grouped by active parameters (1B, 3B, 7-8B, 13-14B, 27-34B) including recent models like Llama3.2, Gemma2, Qwen2, and Mistral to assess the efficiency of MoE architectures enhanced with RoMA.

**Training Set** comprises 49,000 samples distributed across five task categories, as shown in Figure 4. The dataset includes General Knowledge tasks (BIG-Bench and Super-GLUE), Commonsense reasoning (CommonsenseQA and SocialIQA), Science QA (OpenBookQA and SciQ), Reading comprehension (MultiRC), and Coreference resolution (KnowRef). This diverse composition ensures comprehensive coverage across different reasoning capabilities for effective training.

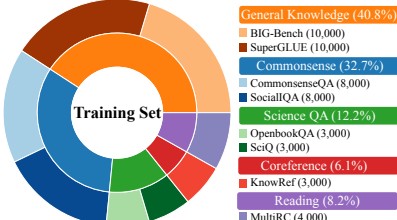

Figure 4: Training set statistics.

**Benchmarks** We evaluate RoMA on eight diverse benchmarks. The evaluation suite includes MMLU, HellaSwag, PIQA, ARC-Challenge, ARC-Easy, WinoGrande, BoolQ and GSM8K. Notably, GSM8K serves as an Out-Of-Distribution (OOD) benchmark since our training set doesn't contain math-related data. Details about training set and benchmarks is in Appendix A.8 and A.9.

### 5.2 MAIN RESULTS

**Advantage of RoMA over different adaptation methods.** Table 1 compares adaptation methods on OLMoE, DeepSeekMoE, and Qwen3-MoE across eight benchmarks. Lightweight methods (ICL, Router/Prefix/Prompt Tuning) yield only modest gains, while Oracle tuning and Dense BP achieve stronger, though still limited, improvements relative to the Oracle upper bound. C3PO performs better than these baselines, yet RoMA achieves the highest overall accuracy. On MMLU, RoMA boosts DeepSeekMoE from 46.2% to 56.8% (+10.6%) and OLMoE from 57.8% to 69.0% (+11.2%), surpassing C3PO by +1.4% and +3.5%, respectively. Although C3PO achieves comparable accuracy as RoMA, its inference cost is 6–7× higher than both RoMA and the base model (See Figure 5), highlighting RoMA 's superior efficiency–effectiveness trade-off. In addition, RoMA shows more advantages over C3PO on larger models such as DeepSeekMoE and Qwen3-MoE. The accuracy and cost of the other two models are reported in Appendix A.1 and A.2. We further compare RoMA with parameter-efficient fine-tuning (PEFT) methods, including LoRA, DoRA, and MoLE, applied to the router parameters. As detailed in Appendix A.4, RoMA outperforms these methods by significant margins (+7.5% ∼ +8.6% on average) while introducing zero new parameters, highlighting that manifold alignment is more effective than merely increasing parameter capacity for routing optimization.

**Comparison of routing weights manifold before and after RoMA.** Figure 3 illustrates the effect of RoMA on routing weights. After applying RoMA, routing weights form clear clusters (Figure 3(a))

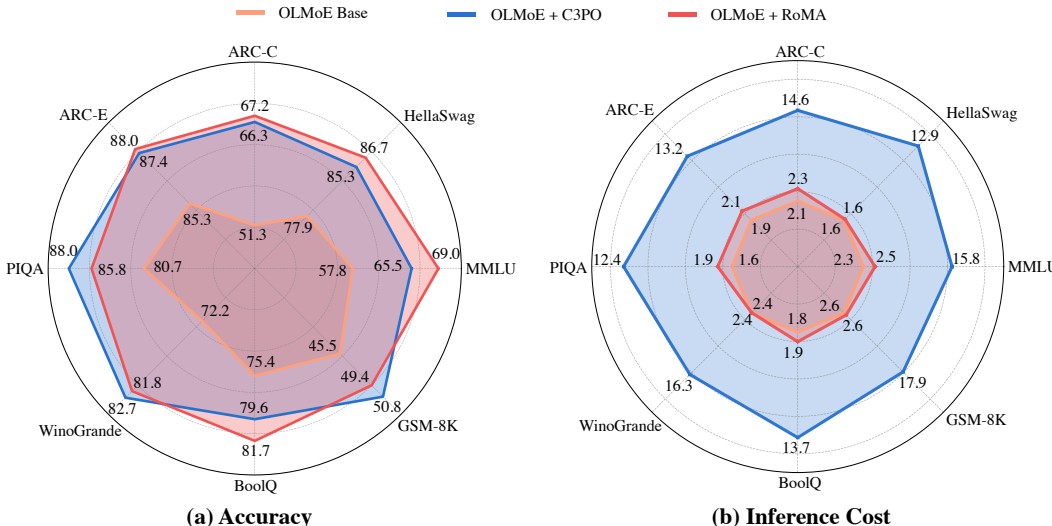

Figure 5: Performance and inference cost of OLMoE (base model), OLMoE + C3PO and OLMoE + RoMA across eight benchmarks. **(a)** Accuracy: RoMA consistently improves the base model's performance to be comparable or better than C3PO. **(b)** Inference cost (in FLOPs $\times 10^{11}$): RoMA maintains nearly the same efficiency as the base model, while C3PO requires test-time optimization and induces 6–7$\times$ more FLOPs. These results highlight the effectiveness and efficiency of RoMA.

that closely align with the task embedding structure (Figure 3(c)). In contrast, the pretrained routing weights show little alignment with task clusters in Figure 3(b), highlighting that RoMA effectively resolves the manifold misalignment problem. Furthermore, the post-RoMA routing patterns closely resemble the oracle routing weights as shown in Figure 3(d), suggesting that our optimization moves the model toward theoretically optimal expert assignments. As a result, samples within the same task cluster receive similar routing patterns, enabling more consistent and efficient use of specialized expertise and bridging the performance gap between suboptimal pretrained routing and ideal oracle routing.

**Advantage of RoMA over State-of-the-Art models.** Table 2 reports LLM performance across eight benchmarks with varying active parameter counts. Notably, OLMoE-7B-A1B+RoMA, with only 1B active parameters, achieves 69.0% on MMLU and 86.7% on HellaSwag, surpassing several 7–8B and even 13B dense models. Similarly, DeepSeekMoE-16B-A3B+RoMA (3B active) delivers substantial gains, matching or exceeding the performance of dense LLMs up to 34B parameters. These results demonstrate that RoMA consistently improves routing quality, enabling small active-parameter MoEs to rival or outperform much larger dense counterparts. Details of models are in Appendix A.10.

## 5.3 ABLATION STUDY

We perform a series of ablation studies to systematically analyze the design choices behind RoMA. Specifically, we investigate: (i) which layers to regularize, (ii) which token positions to use for routing guidance, (iii) how to select neighbors for manifold alignment, (iv) the effect of training set size, and (v) the choice of regularization method. These ablations help identify the most effective and efficient configuration, revealing which factors are critical for performance. All experiments are conducted on OLMoE, and experiment results on DeepSeekMoE are provided in the Appendix A.11.

**Layer Selection** Figure 6 examines how applying routing manifold regularization to different subsets of layers affects model performance. Applying RoMA to a single layer yields only modest gains (69.1–69.7%), while extending it to two layers improves accuracy above 71%. Performance continues to increase as more layers are regularized, with the last five layers (L5) achieving the highest accuracy of 76.2%, even surpassing the All-Layer configuration (75.1%). These results highlight that the final layers are particularly critical for routing quality, and that selectively regularizing a small set of strategically important layers is both more effective and more efficient than uniformly applying RoMA across all layers.

Table 1: Comparison of RoMA with the Base model, Oracle, test-time adaptation methods (ICL, C3PO), training-based methods (Router/Oracle/Prefix/Prompt Tuning), across eight benchmarks on DeepSeekMoE, OLMoE, and Qwen3-30B-A3B. Details of the baselines and benchmarks are provided in Section 5.1. **Bold** numbers denote the best performance (excluding Oracle), and underlined numbers denote the second best. RoMA improves DeepSeekMoE from 46.2% to 56.8% (+10.6%), improves OLMoE from 57.8% to 69.0% (+11.2%), and improves Qwen3-30B-A3B from 74.2% to 78.8% (+4.6%) on MMLU, outperforming C3PO on all three models.

| Method | MMLU | Hella-Swag | ARC-C | ARC-E | PIQA | Wino-Grande | BoolQ | GSM8K | Avg |
|---|---|---|---|---|---|---|---|---|---|
| **DeepSeekMoE-16B-A3B** | | | | | | | | | |
| Base model | 46.2 | 78.0 | 50.3 | 73.8 | 79.9 | 70.1 | 72.3 | 62.2 | 66.6 |
| Oracle | 63.8 | 92.5 | 70.8 | 85.2 | 90.3 | 82.1 | 83.2 | 71.8 | 80.0 |
| ICL | 49.0 | 81.6 | 56.3 | 76.2 | 81.4 | 72.3 | 75.8 | 65.7 | 69.8 |
| C3PO | 55.4 | 85.7 | **61.6** | 80.7 | **85.8** | **77.5** | 78.2 | **68.5** | 74.2 |
| Router Tuning | 49.3 | 81.5 | 57.2 | 76.6 | 82.0 | 73.8 | 74.5 | 64.8 | 70.0 |
| Oracle Tuning | 54.2 | 84.3 | 60.1 | 79.5 | 84.0 | 76.0 | 77.5 | 66.2 | 72.7 |
| Prefix Tuning | 47.8 | 77.9 | 52.4 | 73.8 | 79.2 | 70.3 | 73.1 | 64.8 | 67.4 |
| Prompt Tuning | 49.3 | 78.6 | 55.1 | 74.7 | 80.5 | 72.0 | 74.2 | 65.5 | 68.7 |
| Dense BP | 50.1 | 80.2 | 54.8 | 77.3 | 81.7 | 74.2 | 76.1 | 63.9 | 69.8 |
| RoMA (Ours) | **56.8** | **87.9** | 61.4 | **81.5** | 85.2 | 76.8 | **80.6** | 67.4 | **74.7** |
| **OLMoE-7B-A1B** | | | | | | | | | |
| Base model | 57.8 | 77.9 | 51.3 | 79.8 | 80.7 | 72.2 | 75.4 | 45.5 | 67.6 |
| Oracle | 72.2 | 91.5 | 74.8 | 91.4 | 93.6 | 87.7 | 84.5 | 53.2 | 81.1 |
| ICL | 60.3 | 80.6 | 58.1 | 82.5 | 83.6 | 76.8 | 78.9 | 48.5 | 71.2 |
| C3PO | 65.5 | 85.3 | 66.3 | 87.4 | **88.0** | **82.7** | 79.6 | **50.8** | 75.7 |
| Router Tuning | 63.2 | 81.7 | 62.5 | 83.8 | 80.9 | 75.3 | 77.8 | 47.2 | 71.6 |
| Oracle Tuning | 66.8 | 84.2 | 65.4 | 86.1 | 86.2 | 80.5 | 79.9 | 49.0 | 74.8 |
| Prefix Tuning | 59.3 | 78.2 | 54.5 | 80.4 | 82.1 | 73.5 | 76.8 | 46.7 | 68.9 |
| Prompt Tuning | 59.7 | 79.5 | 55.9 | 81.3 | 82.4 | 74.1 | 77.2 | 47.3 | 69.7 |
| Dense BP | 61.8 | 82.4 | 57.3 | 84.1 | 83.9 | 76.9 | 75.2 | 48.1 | 71.2 |
| RoMA (Ours) | **69.0** | **86.7** | **67.2** | **88.0** | 85.8 | 81.8 | **81.7** | 49.4 | **76.2** |
| **Qwen3-30B-A3B** | | | | | | | | | |
| Base model | 74.2 | 68.5 | 56.8 | 84.3 | 78.5 | 65.2 | 81.3 | 83.4 | 74.0 |
| Oracle | 82.5 | 80.3 | 69.2 | 92.6 | 87.4 | 77.3 | 90.5 | 90.9 | 83.8 |
| ICL | 75.8 | 70.7 | 59.3 | 86.1 | 80.2 | 67.8 | 83.5 | 84.7 | 76.0 |
| C3PO | 77.9 | 74.1 | 63.4 | 88.1 | 81.7 | 71.9 | **85.4** | 86.0 | 78.6 |
| Router Tuning | 75.3 | 70.3 | 60.1 | 85.7 | 79.8 | 68.5 | 82.8 | 84.2 | 75.8 |
| Oracle Tuning | 77.2 | 73.5 | 62.8 | 87.6 | 81.3 | 71.2 | 84.9 | 85.5 | 78.0 |
| Prefix Tuning | 74.5 | 68.9 | 57.9 | 84.8 | 79.1 | 66.3 | 82.1 | 83.8 | 74.7 |
| Prompt Tuning | 75.0 | 69.6 | 58.6 | 85.2 | 79.6 | 67.0 | 82.7 | 84.0 | 75.2 |
| Dense BP | 76.1 | 71.4 | 59.8 | 86.5 | 80.5 | 69.2 | 83.8 | 84.9 | 76.5 |
| RoMA (Ours) | **78.8** | **74.8** | **65.5** | **88.6** | **83.1** | **73.8** | 85.1 | **86.3** | **79.5** |

**Token Selection** Figure 7 presents the effect of different token selection strategies when applying RoMA. Using multiple tokens (e.g., the first three or middle three) provides moderate improvements over the baseline, with last 3 tokens (Last3) reaching 74.5%. Among single-token choices, the last 1 token (Last1) performs best (76.2%), outperforming both the first one token (First1) (71.4%) and the middle one token (Middle1) (69.2%). These results indicate that the final tokens contain richer task-relevant information for guiding expert routing than earlier or middle tokens. Moreover, the superiority of Last1 over Last3 highlights that a single, well-chosen token can be more effective and efficient than aggregating multiple tokens.

Table 2: Comparison of LLMs with varying active parameters (1B, 3B, 7–8B, 13–14B, 27–34B) evaluated on eight benchmarks. MoE models post-trained by RoMA achieve strong performance, surpassing or matching the performance of much larger dense models. For example, OLMoE-7B-A1B (1B active) achieves 69.0% on MMLU and 86.7% on HellaSwag, outperforming several 7–8B and even 13B dense counterparts. Qwen3-30B-A3B (3B active) achieves 78.8% on MMLU, surpassing even 27–34B dense models, highlighting the effectiveness of MoE+RoMA.

| | MMLU | Hella-Swag | ARC-C | ARC-E | PIQA | Wino-Grande | BoolQ | GSM8K |
|---|---|---|---|---|---|---|---|---|
| **~1B active parameters** | | | | | | | | |
| Llama3.2-1B | 27.4 | 57.9 | 32.1 | 53.9 | 72.4 | 57.4 | 63.7 | 39.4 |
| OLMo-1B | 24.1 | 61.8 | 29.6 | 55.7 | 75.6 | 56.8 | 64.2 | 28.5 |
| OLMoE-7B-A1B | **57.8** | **77.9** | **51.3** | **79.8** | **80.7** | **72.2** | **75.4** | **45.5** |
| **~3B active parameters** | | | | | | | | |
| Gemma2-3B | 43.7 | 66.3 | **58.4** | 75.2 | 71.8 | 64.5 | 73.1 | 41.4 |
| DeepSeekMoE-16B-A3B | 46.2 | **78.0** | 50.3 | 73.8 | **79.9** | **70.1** | 72.3 | 62.2 |
| Qwen3-30B-A3B | **74.2** | 68.5 | 56.8 | **84.3** | 78.5 | 65.2 | **81.3** | **83.4** |
| **~7-8B active parameters** | | | | | | | | |
| Qwen2-7B | 53.4 | 74.9 | 45.8 | 69.7 | 77.2 | 68.1 | **84.8** | **79.9** |
| Mistral-7B | **59.6** | **81.0** | **53.8** | 79.6 | **82.2** | **74.0** | 68.1 | 37.9 |
| Llama3.1-8B | 57.7 | 77.9 | 48.7 | **80.8** | 81.4 | 73.5 | 81.9 | 49.6 |
| **~13-14B active parameters** | | | | | | | | |
| Llama2-13B | 53.8 | 78.6 | 50.1 | 74.5 | 79.1 | 70.1 | 75.7 | 35.2 |
| Vicuna-13B | 51.3 | 76.2 | 47.4 | 72.8 | 78.0 | 68.2 | 71.5 | 32.2 |
| Qwen1.5-14B | **66.7** | **81.5** | **58.0** | **85.3** | **82.1** | **76.9** | **81.3** | **58.4** |
| **~27-34B active parameters** | | | | | | | | |
| Gemma2-27B | **75.2** | **86.4** | **71.4** | **88.9** | **83.2** | **79.0** | **84.5** | 61.3 |
| Yi-34B | 73.5 | 83.1 | 58.2 | 82.6 | 82.6 | 78.9 | 83.1 | **63.5** |
| Llama2-34B | 62.6 | 79.4 | 54.5 | 77.5 | 81.9 | 76.0 | 78.1 | 42.2 |
| **RoMA (Ours)** | | | | | | | | |
| DeepSeekMoE-16B-A3B | 56.8 | **87.9** | 61.4 | 81.5 | 85.2 | 76.8 | 80.6 | 67.4 |
| OLMoE-7B-A1B | 69.0 | 86.7 | **67.2** | 88.0 | **85.8** | **81.8** | 81.7 | 49.4 |
| Qwen3-30B-A3B | **78.8** | 74.8 | 65.5 | **88.6** | 83.1 | 73.8 | **85.1** | **86.3** |

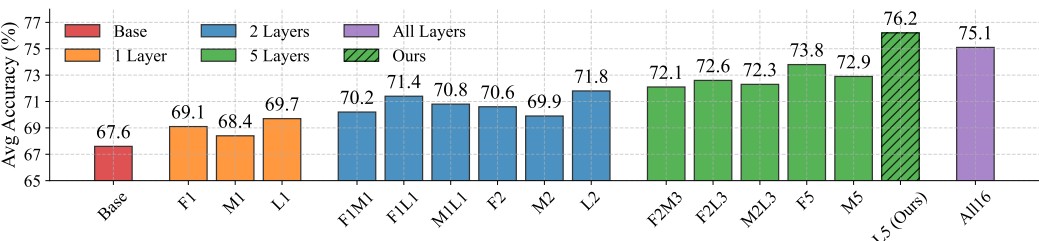

Figure 6: Applying RoMA at different layers (F: early layers, M: middle layers, L: late layers). Fine-tuning the routers in the last five layers (L5, RoMA) achieves the best performance.

**Neighborhood Selection** Figure 8 compares different strategies for selecting neighbors in RoMA. Random neighbor selection yields almost no improvement over the baseline (67.8% vs. 67.6%). Using $\epsilon$-neighborhoods shows sensitivity to the choice of radius: performance improves steadily from 68.9% ($\epsilon$=0.3) to a peak of 74.1% at $\epsilon$=0.5, but drops slightly when the radius grows larger ($\epsilon$=0.7). In contrast, $k$-nearest neighbor selection provides more stable gains, with $k$=3 achieving the best overall accuracy of 76.2%. Notably, this surpasses both smaller ($k$=1) and larger ($k$=5) settings, suggesting that a moderate number of neighbors balances robustness and noise. These results

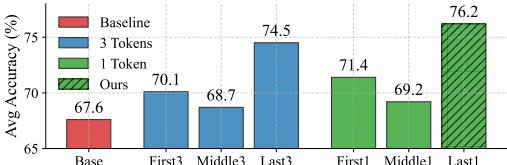

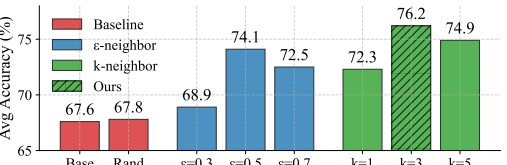

Figure 7: Applying RoMA to routing weights of tokens at different positions. Regularizing the Last1 token's routing weights performs the best.

Figure 8: Comparing neighbor selection strategies in RoMA. Rand—random neighbors. $k$-NN with $k = 3$ achieves the best performance.

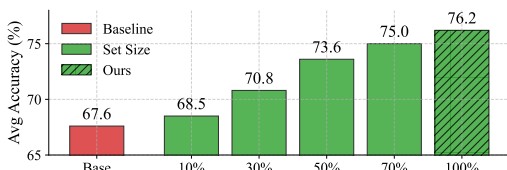

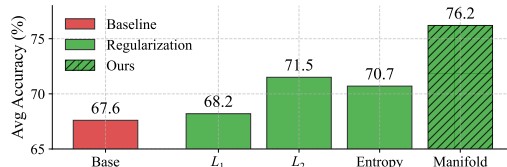

Figure 9: Comparing different training set sizes for RoMA on OLMoE. While the full training set (100%) yields the best performance, 30% suffices to achieve substantial gains over the baselines.

Figure 10: Comparing different regularization methods with RoMA. RoMA's manifold regularization achieves the best performance.

highlight that careful neighborhood design is crucial for effective manifold alignment, and that our chosen $k$=3 strategy offers the most reliable improvement.

**Training Set Size** Figure 9 examines how the size of the training set used for RoMA affects performance. Starting from the baseline accuracy of 67.6%, even using only 10% of the training data yields a noticeable gain (68.5%). Performance improves steadily as more data is available, reaching 70.8% at 30% and 73.6% at 50%. With 70% of the data, accuracy rises further to 75.0%, and using the full dataset achieves the best performance of 76.2%. These results demonstrate that RoMA benefits consistently from additional training data, but also that substantial improvements can already be obtained with a relatively small fraction of the dataset, highlighting its data efficiency.

**Regularization Methods** Figure 10 compares different regularization strategies applied to the router. Standard techniques such as $L_1$ and $L_2$ penalties yield only modest improvements over the baseline (68.2% and 71.5%, respectively), while entropy regularization reaches a similar level (70.7%). In contrast, our proposed manifold regularization achieves the best result of 76.2%, substantially outperforming all alternatives. This demonstrates that aligning routing weights in the task embedding space provides a more effective inductive bias than generic sparsity or entropy-based constraints, highlighting the unique advantage of RoMA's manifold perspective.

**Embedding Models** We further investigate the sensitivity of RoMA to the choice of embedding models. We evaluate RoMA using diverse embedding models ranging from 22M to 7.8B parameters (including all-MiniLM, BGE, and Qwen-embedding). As detailed in Appendix A.7, RoMA achieves consistent accuracy improvements ($+3.6\% \sim +8.6\%$) across all tested models.

# 6 CONCLUSIONS

Our work introduces RoMA, a lightweight router post-training method for sparse Mixture-of-Experts LLMs. By aligning routing weights with the underlying task embedding manifold through manifold regularization, RoMA addresses the fundamental misalignment between task understanding and expert utilization in MoE models. Our approach requires updating only router parameters while keeping experts frozen, yet consistently improves accuracy across diverse benchmarks by 7–15% without increasing inference cost. Extensive experiments demonstrate that RoMA enables small active-parameter MoEs to rival or even surpass much larger dense models, highlighting both the efficiency and effectiveness of RoMA. Beyond performance gains, our findings underscore the importance of geometric alignment between task representation and expert selection, offering new insights for advancing routing strategies in future MoE architectures.

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

# A APPENDIX

## A.1 COMPARISON BETWEEN ROMA AND BASELINES ON DEEPSEEKMOE-16B-A3B AND QWEN3-30B-A3B

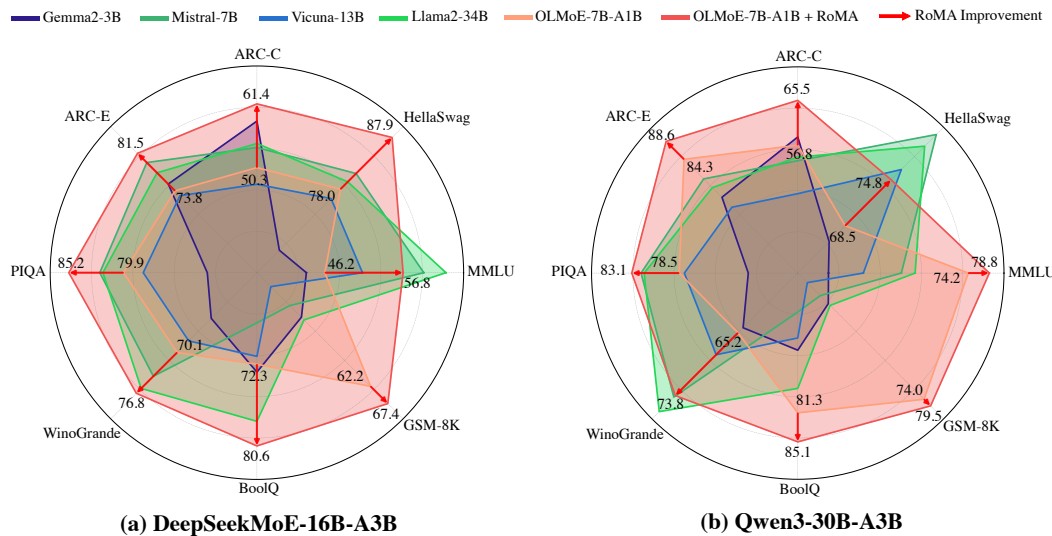

(a) DeepSeekMoE-16B-A3B

(b) Qwen3-30B-A3B

Figure 11: **(a)**: Radar figure of DeepSeekMoE-16B-A3B, **(b)** Radar figure of Qwen3-30B-A3B. RoMA consistently improves model's performance on multiple benchmarks.

## A.2 ACCURACY AND COST OF DEEPSEEKMOE-16B-A3B AND QWEN3-30B-A3B

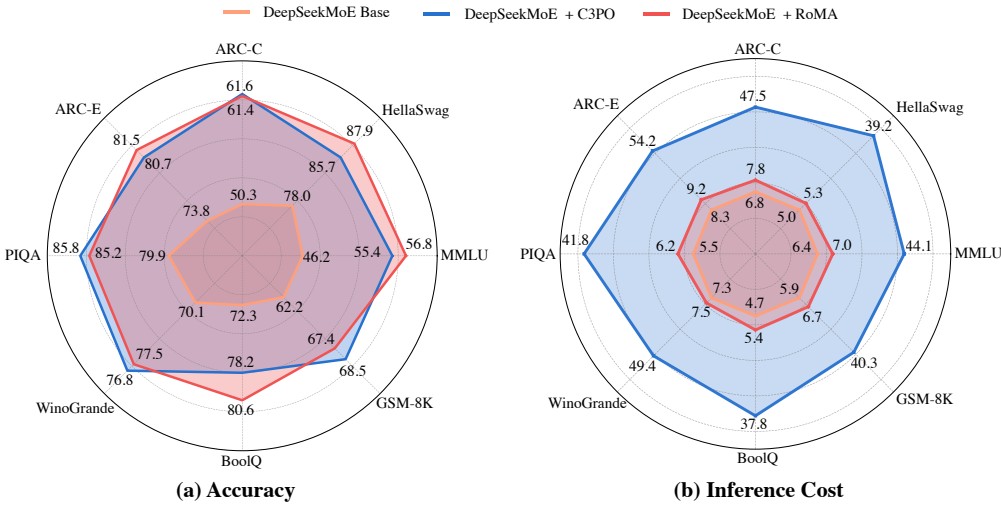

(a) Accuracy

(b) Inference Cost

Figure 12: **(a)** Accuracy: RoMA achieves similar accuracy improvement as C3PO on DeepSeekMoE-16B-A3B. **(b)** Inference cost (in FLOPs $\times 10^{11}$): RoMA maintains nearly the same efficiency as the base model, while C3PO requires test-time optimization and induces 6–7$\times$ more FLOPs.

## A.3 QUANTITATIVE ALIGNMENT ANALYSIS

To supplement the UMAP visualizations in Figure 3, we conduct quantitative alignment analyses on ARC-C and MMLU benchmarks. We report three metrics: (1) **Subspace Similarity (CKA)** to

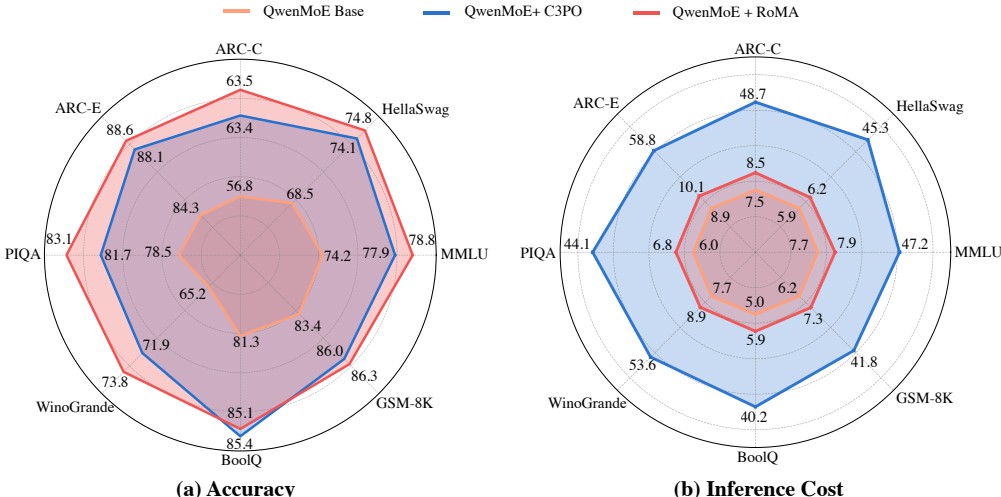

**(a) Accuracy**  **(b) Inference Cost**

Figure 13: **(a)** Accuracy: RoMA achieves similar accuracy improvement as C3PO on Qwen3-30B-A3B. **(b)** Inference cost (in FLOPs $\times 10^{11}$): RoMA maintains nearly the same efficiency as the base model, while C3PO requires test-time optimization and induces 6–7$\times$ more FLOPs.

measure the similarity between routing-weight subspaces and task embeddings; (2) **k-NN Neighbor Consistency** ($k = 5$) to measure how well routing decisions preserve semantic neighborhoods; and (3) **Trustworthiness Score** ($k = 10$) to evaluate local structure preservation.

Table 3: Quantitative alignment metrics before and after RoMA. All metrics show significant improvement, quantitatively confirming the manifold alignment.

| Dataset | CKA Similarity (↑) | | k-NN Consistency (↑) | | Trustworthiness (↑) | |
|---|---|---|---|---|---|---|
| | Base | RoMA | Base | RoMA | Base | RoMA |
| ARC-C | 0.18 | **0.47** | 24.3% | **51.7%** | 0.53 | **0.76** |
| MMLU | 0.21 | **0.52** | 26.8% | **54.2%** | 0.56 | **0.79** |

## A.4 COMPARISON WITH PEFT METHODS

We compare RoMA with representative PEFT baselines (LoRA, DoRA, MoLE) applied to the router parameters. The results are averaged over 8 benchmarks. As shown in Table 4, RoMA achieves superior performance without introducing any new parameters during inference.

Table 4: Comparison with PEFT methods on OLMoE-7B-A1B (Last 5 Layers). RoMA outperforms PEFT variants with fewer trainable parameters and 0 new parameters.

| Method | Rank | Trainable Params | New Params | Avg. Acc (%) | △ Acc |
|---|---|---|---|---|---|
| Baseline | – | – | – | 67.6 | – |
| LoRA | 16 | 331K | 331K | 71.2 | +3.6 |
| DoRA | 16 | 334K | 334K | 71.6 | +4.0 |
| MoLE (3×LoRA) | 16 | 1.00M | 1.00M | 72.5 | +4.9 |
| **RoMA (Ours)** | – | 658K | **0** | **76.2** | **+8.6** |

## A.5 TRAINING COST ANALYSIS

We provide a breakdown of the training cost for RoMA on OLMoE-7B-A1B (49K samples). The training process consists of task embedding computation, k-NN search, and router fine-tuning.

Table 5: FLOPs breakdown for RoMA training. The k-NN search cost is negligible.

| Component | FLOPs ($\times 10^{15}$) | Share |
|---|---|---|
| Task Embedding (Pre-computed) | 3.8 | 3.0% |
| k-NN Search (FAISS) | 0.002 | $< 0.01\%$ |
| Router Fine-tuning | 122 | 97.0% |
| **Total** | **125.8** | **100%** |

As shown in Table 5, the overhead from k-NN search is negligible ($< 0.01\%$) thanks to efficient approximate nearest neighbor search (FAISS). The task embeddings are computed once and reused.

## A.6 BIAS MITIGATION WITH CURRICULUM LEARNING

To address the concern that imitating only "successful neighbors" might introduce confirmation bias, we explored a curriculum learning strategy. We tested a 3-stage curriculum: (1) Early stage: include neighbors from both correct and incorrect predictions (top-30% similarity); (2) Mid stage: top-50% similarity mixed; (3) Late stage: strict filter (only correct predictions).

The curriculum strategy yields a slight improvement (76.4%) over the strict filter (76.2%), while an always-soft filter degrades performance (74.1%). This suggests that while a strict filter is robust, relaxing it early in training can provide marginal benefits. For simplicity and robustness, the main results in the paper use the strict filter.

## A.7 ROBUSTNESS ACROSS EMBEDDING MODELS

We evaluated RoMA on OLMoE-7B-A1B using various embedding models to assess its sensitivity. The models include all-MiniLM-L6-v2 (22M), all-mpnet-base-v2 (110M), Qwen3-0.6B-embedding (0.6B), bge-multilingual-gemma2 (2.6B), gte-Qwen2-7B-instruct (7B), and NV-Embed-v2 (7.8B).

Table 6: Performance of RoMA with different embedding models on OLMoE-7B-A1B. RoMA brings consistent improvements across varying embedding model sizes.

| Embedding Model | Size | Avg. Acc (%) | $\Delta$ Acc |
|---|---|---|---|
| Baseline (OLMoE) | – | 67.6 | – |
| all-MiniLM-L6-v2 | 22M | 71.2 | +3.6 |
| all-mpnet-base-v2 | 110M | 72.5 | +4.9 |
| Qwen3-0.6B-embedding | 0.6B | 75.9 | +8.3 |
| bge-multilingual-gemma2 | 2.6B | 73.4 | +5.8 |
| gte-Qwen2-7B-instruct | 7B | 75.1 | +7.5 |
| **NV-Embed-v2 (Ours)** | 7.8B | **76.2** | **+8.6** |

As shown in Table 6, RoMA consistently improves performance regardless of the embedding model size. Larger embedding models generally bring better alignment (higher accuracy gain), but even compact models provide meaningful improvements.

## A.8 DETAILS OF TRAINING SET

Our training set comprises 49,000 samples distributed across five task categories, ensuring comprehensive coverage across diverse reasoning skills:

- **BIG-Bench** (Wei et al., 2022): A large-scale collaborative benchmark covering a wide range of tasks such as logical reasoning, linguistic phenomena, and commonsense knowledge. It is designed to probe broad generalization and emergent capabilities in large language models.
- **SuperGLUE** (Wang et al., 2019): A benchmark suite for general natural language understanding, consisting of challenging tasks such as natural language inference, word sense disambiguation, and question answering. It extends the original GLUE benchmark to push models toward higher-level reasoning.

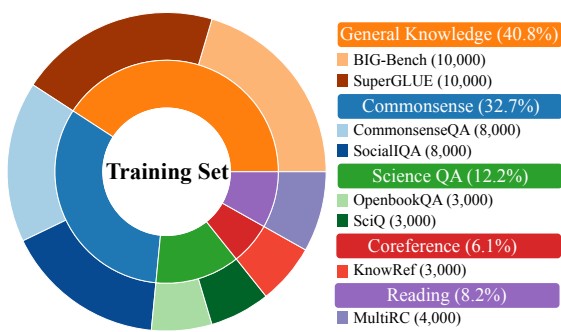

Figure 14: Training Set by Task

| Benchmarks | Size |
| --- | --- |
| MMLU | 14,042 |
| HellaSwag | 10,042 |
| PIQA | 1,838 |
| ARC-C | 1,172 |
| ARC-E | 2,376 |
| WinoGrande | 1,267 |
| BoolQ | 3,227 |
| GSM8k | 1,000 |

Figure 15: Overview of evaluation benchmarks we use for router post-training. Benchmarks are reserved strictly for evaluation.

- **CommonsenseQA** (Talmor et al., 2019): A multiple-choice dataset focusing on commonsense reasoning, requiring models to connect concepts and apply everyday knowledge.
- **SocialIQA** (Sap et al., 2019): A benchmark for social commonsense reasoning, where models must infer likely intents, reactions, and motivations of people in everyday situations.
- **OpenBookQA** (Mihaylov et al., 2018): A science-oriented QA dataset requiring both retrieval from a small "open book" of facts and additional commonsense reasoning.
- **SciQ** (Welbl et al., 2017): A dataset of science exam-style questions covering physics, biology, and chemistry, testing factual recall and reasoning in scientific contexts.
- **MultiRC** (Khashabi et al., 2018): A reading comprehension benchmark with multi-sentence passages and multi-answer questions, requiring deeper reasoning across long contexts.
- **KnowRef** (Emami et al., 2019): A coreference resolution dataset where multiple entities are mentioned, and models must resolve ambiguous pronouns using contextual cues.

## A.9 DETAILS OF BENCHMARKS

We evaluate our method on eight widely used benchmarks covering general knowledge, commonsense reasoning, science QA, and mathematical problem-solving:

- **MMLU** (Hendrycks et al., 2021): A comprehensive benchmark of 57 subjects spanning STEM, humanities, social sciences, and professional domains. It measures models' multitask accuracy and general world knowledge.
- **HellaSwag** (Zellers et al., 2019): A commonsense benchmark requiring models to select the most plausible continuation of a given context. It emphasizes grounded reasoning about everyday scenarios.
- **PIQA** (Bisk et al., 2020): A physical commonsense reasoning dataset, where models must infer the correct solution to physical problems from everyday settings.
- **ARC-Challenge** (Clark et al., 2018): A benchmark of challenging grade-school science questions requiring reasoning, knowledge retrieval, and integration across multiple facts.
- **ARC-Easy** (Clark et al., 2018): The easier subset of ARC focusing on factual recall and simpler reasoning in grade-school science.
- **WinoGrande** (Sakaguchi et al., 2020): A large-scale benchmark for pronoun resolution and commonsense reasoning, designed to be adversarially filtered and less susceptible to dataset artifacts.
- **BoolQ** (Clark et al., 2019): A reading comprehension dataset of yes/no questions paired with passages from Wikipedia, requiring models to integrate text understanding with factual reasoning.
- **GSM8k** (Cobbe et al., 2021): A dataset of grade-school math word problems requiring multi-step numerical reasoning. We treat GSM8k as an out-of-distribution (OOD) evaluation since math is not included in our training set.

## A.10 MODEL DESCRIPTIONS

In this section, we provide additional details of the models reported in Table 2. These models cover a broad range of active parameters (1B, 3B, 7–8B, 13–14B, 27–34B), including dense LLMs and sparse Mixture-of-Experts (MoE) variants. All results are reported in the main text (see Table 2).

**Models with ∼1B Active Parameters**

- Llama3.2-1B (Dubey et al., 2024): A 1B-parameter dense model in the Llama 3.2 family.
- OLMo-1B (Groeneveld et al., 2024): Dense model from the AllenAI OLMo family.
- OLMoE-7B-A1B Muennighoff et al. (2024): A sparse MoE variant of OLMoE with 7B total parameters and ∼1B active per token.

**Models with ∼3B Active Parameters**

- Gemma2-3B (Team, 2024a): Google's Gemma2 family dense model.
- Qwen1.5-14B-A3B (Bai et al., 2023): Sparse MoE variant of Qwen1.5 with 14B total parameters and ∼3B active per token.
- DeepSeekMoE-16B-A3B (Dai et al., 2024b): Sparse MoE model with 16B total parameters and ∼3B active per token.

**Models with ∼7–8B Active Parameters**

- Qwen2-7B (Team, 2024b): Dense model from the Qwen2 family.
- Mistral-7B (Jiang et al., 2023): Dense model emphasizing efficient training and inference.
- Llama3.1-8B (Dubey et al., 2024): A dense model from the Llama 3.1 family.

**Models with ∼13–14B Active Parameters**

- Llama2-13B (Touvron et al., 2023): Dense model from the Llama 2 release.
- Vicuna-13B (Chiang et al., 2023): Instruction-tuned LLM based on Llama2-13B.
- Qwen1.5-14B (Bai et al., 2023): Dense version of Qwen1.5.

**Models with ∼27–34B Active Parameters**

- Gemma2-27B (Team, 2024a): Largest Gemma2 dense variant.
- Yi-34B (Young et al., 2024): Dense model with 34B parameters.
- Llama2-34B (Touvron et al., 2023): Large dense model from the Llama 2 family.

## A.11 DETAILS OF ABLATION STUDY ON DEEPSEEKMOE

In this section, we provide a detailed ablation study of RoMA on the DeepSeekMoE model.

**Layer configurations (Table 7).** When applying RoMA to different numbers of layers, we find that using a single layer or two layers provides modest gains (e.g., F1 at 68.0%, L2 at 70.6%). Increasing to five layers yields further improvements (up to 72.4%), while applying RoMA on all layers gives 73.7%. Interestingly, restricting the method to the last five layers achieves the best result (74.7%), surpassing even the all-layer setting, which highlights the importance of critical-layer selection.

**Token positions (Table 8).** We next evaluate applying RoMA to the routing weights of different tokens. Regularizing the first or middle tokens shows only limited improvements (69.0% and 67.6%, respectively), while the last token positions provide stronger performance. In particular, Last1 achieves the best result (74.7%), indicating that the most informative supervision signal for routing weights lies in the final tokens.

Table 7: DeepSeekMoE average accuracy (%) across different layer configurations

| Group | Configuration | Avg. Acc. (%) |
|---|---|---|
| Base | Base | 66.6 |
| 1 Layer | F1 | 68.0 |
| | M1 | 67.4 |
| | L1 | 68.6 |
| 2 Layers | F1M1 | 69.0 |
| | F1L1 | 70.2 |
| | M1L1 | 69.6 |
| | F2 | 69.4 |
| | M2 | 68.8 |
| | L2 | 70.6 |
| 5 Layers | F2M3 | 70.8 |
| | F2L3 | 71.3 |
| | M2L3 | 71.0 |
| | F5 | 72.4 |
| | M5 | 71.6 |
| All Layers | All16 | 73.7 |
| Ours | L5 (Ours) | **74.7** |

Table 8: DeepSeekMoE average accuracy (%) when applying RoMA on different token positions.

| Group | Configuration | Avg. Acc. (%) |
|---|---|---|
| Baseline | Base | 66.6 |
| 3 Tokens | First3 | 69.0 |
| | Middle3 | 67.6 |
| | Last3 | 73.1 |
| 1 Token | First1 | 70.2 |
| | Middle1 | 68.1 |
| Ours | Last1 (Ours) | **74.7** |

Table 9: DeepSeekMoE average accuracy (%) with different neighbor selection strategies.

| Group | Configuration | Avg. Acc. (%) |
|---|---|---|
| Baseline | Base | 66.6 |
| Baseline | Rand | 66.8 |
| $\varepsilon$-neighbor | $\varepsilon = 0.3$ | 68.0 |
| | $\varepsilon = 0.5$ | 72.8 |
| | $\varepsilon = 0.7$ | 71.2 |
| k-neighbor | k=1 | 71.0 |
| | k=3 (Ours) | **74.7** |
| | k=5 | 73.4 |

**Neighbor selection (Table 9).** We compare $\varepsilon$-neighbor and $k$-neighbor strategies. Small $\varepsilon$ (0.3) gives a minor improvement (68.0%), while moderate $\varepsilon$ (0.5) achieves a strong 72.8%. For $k$-neighbors, increasing $k$ improves accuracy up to $k = 3$ (74.7%), after which performance saturates ($k = 5$ at 73.4%). This suggests that a balanced neighbor selection (neither too sparse nor too dense) is crucial for generalization.

**Training set size (Table 10).** We investigate different proportions of training data used for regularization. Performance grows steadily with larger set sizes, from 67.5% at 10% to 73.6% at 70%. Using the full dataset (100%) achieves the best result (74.7%), confirming that more data consistently strengthens the alignment of routing weights.

**Regularization methods (Table 11).** We compare different regularization objectives. $L_1$ and $L_2$ losses improve the baseline to 67.2% and 70.3%, respectively, while entropy regularization achieves 69.8%. Our proposed manifold regularization significantly outperforms all alternatives, reaching 74.7%, which demonstrates the effectiveness of aligning routing weights with the manifold structure of successful neighbors.

Table 10: DeepSeekMoE average accuracy (%) with different training set sizes.

| Configuration | Avg. Acc. (%) |
|---|---|
| Base | 66.6 |
| 10% | 67.5 |
| 30% | 69.8 |
| 50% | 72.2 |
| 70% | 73.6 |
| 100% (Ours) | **74.7** |

Table 11: DeepSeekMoE average accuracy (%) with different regularization methods.

| Configuration | Avg. Acc. (%) |
|---|---|
| Base | 66.6 |
| $L_1$ | 67.2 |
| $L_2$ | 70.3 |
| Entropy | 69.8 |
| Manifold (Ours) | **74.7** |

## THE USE OF LARGE LANGUAGE MODELS (LLMS)

We used large language models (LLMs) solely for grammar checking and language polishing. No part of the research design, methodology, experimental results, or core writing was generated by LLMs. All scientific ideas, analyses, and conclusions are entirely the work of the authors.

