# OpenReview forum: "Routing Manifold Alignment Improves Generalization of Mixture-of-Experts LLMs"
_ICLR.cc/2026/Conference — ICLR 2026 Poster_

### Official Review · Reviewer_8yaH · 2025-10-17

**Soundness:** 3
**Presentation:** 3
**Contribution:** 3
**Rating:** 8
**Confidence:** 3

**Summary:**

This paper proposes a novel post-training method, termed Routing Manifold Alignment (RoMA), that fine-tunes routing parameters to address the misalignment between routing weights and task semantics in MoE models. The authors first analyze the mismatch between routing selection and tasks in existing MoE models. Based on this, RoMA aligns the manifold of task and routing. Sufficient experiments have demonstrated the effectiveness of the proposed method.

**Strengths:**

1. The proposed post-routing training idea of ​​explicitly aligning the "task embedding manifold" with the "routing weight manifold" is very novel.

2. Experiments have fully demonstrated the effectiveness of the proposed method and achieved significant improvements on different datasets.

**Weaknesses:**

1. The proposed method requires an additional embedding model to provide annotations, and its performance may be limited by the additional model.

2. Further quantitative analysis is needed to illustrate the impact of the proposed method. (detailed in questions)

**Questions:**

1. Although it achieves good performance on a range of downstream tasks, does this adjustment to the expert distribution compromise the model's expressive power?

2. The authors need to provide some theoretical analysis of the ablation study. The current analysis is limited to experimental phenomena, and the authors should provide some theoretical analysis of the causes.

---

> ### Author Response · Authors · 2025-11-21
> **Response to Reviewer 8yaH**
>
> Thank you for your detailed feedback! We address your comments below.
>
> > **W1. The proposed method requires an additional embedding model to provide annotations, and its performance may be limited by the additional model.**
>
> Thank you for the question. As shown in `General Question Q2`, RoMA achieves **consistent gains across diverse embedding models**, indicating robustness to the choice of embedding model.
> Moving forward, we plan to (1) evaluate more **SOTA embedding models** for further improvement, and (2) explore using the **MoE model itself as the embedding model** [1], enabling a fully self-contained routing enhancement.
>
> [1] Your Mixture-of-Experts LLM Is Secretly an Embedding Model For Free (Li et al, ICLR 2025)
>
> > **W2 & Q1. Further quantitative analysis is needed to illustrate the impact of the proposed method, does this adjustment to the expert distribution compromise the model's expressive power?**
>
> Thank you for the question. **RoMA only fine-tunes router parameters while keeping all experts frozen**, thereby **preserving the model’s full representational capacity**. Since expressive power primarily arises from the experts rather than the routing mechanism, the **fundamental capacity of the MoE model remains intact**.
>
> To further verify this, we evaluated RoMA on **MT-Bench** [1], which assesses expressive capability through nuanced, multi-turn generation. MT-Bench serves as an OOD benchmark since our training set doesn't contain multi-turn dialogue data. The results demonstrate that RoMA not only preserves but enhances expressive power, achieving improvements.
>
> | Model                      | MT-Bench Score | Δ         |
> | -------------------------- | -------------- | --------- |
> | OLMoE-7B-A1B       		 | 6.21       | –         |
> | OLMoE-7B-A1B + RoMA        | 6.40       | +0.19 |
> | DeepSeekMoE-16B-A3B 		 | 7.21       | –         |
> | DeepSeekMoE-16B-A3B + RoMA | 7.24       | +0.03 |
>
>
> [1] MT-Bench-101: A Fine-Grained Benchmark for Evaluating Large Language Models in Multi-Turn Dialogues (Bai et al, ACL 2024)
>
> > **Q2. The current analysis of ablation study is limited to experimental phenomena, and the authors should provide some theoretical analysis of the causes.**
>
> Thank you for the question. Below we provide theoretical insights into our ablation results:
>
> (1) **Last 5 Layers:**
> Deeper layers encode higher-level task semantics with lower intrinsic dimensionality and clearer manifold structure. Thus, manifold alignment is most effective in these layers where neighborhood relations are reliable.
>
> (2) **Last Token:**
> As discussed in `General Question Q1`, the last token aggregates contextual information, making it ideal for sample-level representation.
>
> (3) **k = 3:**
> This reflects a bias–variance tradeoff: small *k* (e.g., 1) is noise-sensitive, while large *k* (e.g., 5) introduces irrelevant neighbors. *k = 3* achieves a robust local balance.
>
> (4) **Manifold Regularization:**
> Unlike L1/L2 penalties that ignore geometry, manifold regularization enforces **local consistency**—similar tasks in embedding space share similar routing patterns—enhancing transfer and generalization.

---

### Official Review · Reviewer_yq6K · 2025-10-26

**Soundness:** 1
**Presentation:** 2
**Contribution:** 2
**Rating:** 2
**Confidence:** 4

**Summary:**

The paper proposes RoMA (Routing Manifold Alignment), a post-training method for Mixture-of-Experts (MoE) LLMs that updates only the routers. The idea is to build a neighbor graph in a task-embedding space over a subset of “successful” training examples (those the current model gets right), then add a *manifold regularizer* that encourages a sample’s routing pattern ($r_i$) to resemble those of its successful neighbors ($r_j$), with similarity-normalized weights ($W_{i,j}$). The method adds no inference-time cost and is reported to improve accuracy across benchmarks on MoE backbones. The paper’s motivation leans on a claimed gap to an “oracle routing” upper bound and on UMAP visualizations of routing vs task-embedding manifolds.

**Strengths:**

- Clear, practical objective: The manifold regularizer is standard, well-posed, and easy to implement for router-only fine-tuning.
- **Empirical breadth:** The paper reports consistent gains over a wide suite (MMLU, ARC-C/E, HellaSwag, etc.) and shows competitive performance relative to a strong test-time routing method (C3PO).
- **Focused ablations:** The paper varies the number of routed layers, token choice, and neighborhood size, which are some of the design levers that plausibly matter for this approach (though see “Weaknesses” on explanations/variance).
- Writing is generally clear at the high level; figures are visually appealing. The method section is readable.

**Weaknesses:**

**Related works section is really sparse.**
A huge body of related works are present in the literature that are missing and many works such as Moefication etc. are relevant.

**Oracle” routing is defined but not computed, yet used as an empirical anchor.**
The paper defines a per-example oracle  $r_i^*=\arg\min_r \mathcal L_{\mathrm{CE}}(f(x_i,r),y_i)$ to claim a 10–20% “oracle gap,” and it reports Oracle rows in Table 1. However, no procedure is provided for obtaining these oracle numbers (search/relaxation/heuristic/optimization), nor is the compute or fairness (e.g., use of labels at evaluation) specified. As written, the “oracle” is kind of like a tautological bound, not really a reproducible baseline. Using it in tables materially affects the paper’s narrative but is methodologically unclear.

**Heavy reliance on UMAP visuals without quantitative alignment metrics**
Figure 3 is central to the paper’s motivation (misalignment before RoMA, alignment after, similarity to “oracle” etc.), but UMAP is a non-linear, hyperparameter-sensitive projection, and the paper provides *no quantitative* alignment measures nor any robustness checks across UMAP settings/datasets. As such, the figure is suggestive but insufficient as primary evidence.

**Incomplete reproducibility for key components**
Important knobs/levers are underspecified in the main text:
- **Embedding model ($E(\cdot$)):** name/version, what text is embedded (task descriptions vs inputs), metric/σ and sensitivity.
- **Neighbor graph schedule:** fixed vs recomputed during training; frequency/criteria.
- **Router differentiability:** how gradients traverse the top-(k) gating (STE, Gumbel-Softmax, soft relaxation), and whether training uses hard or soft routing during FT.
- **Hyper-parameters & tuning protocol:** $\lambda$ for the regularizer, $\sigma$, ($k/\epsilon$ seems to be provided in the appendix)..
Without these, independent reproduction will be difficult and also makes the method incredibly opaque and the results hard to interpret.

*Successful neighbors* may introduce *confirmation bias*
The method imitates routing only from samples the current model already solves correctly ((\mathcal S)). This is a plausible heuristic, but it risks **locking in** existing modes and under-serving hard/rare cases; the paper does not probe this trade-off. Can the authors comment on this?

**Statistical rigor and significance**
Main tables lack error bars and multiple seeds. Router-only fine-tuning and MoE routing can be high-variance and single-run improvements, especially single-digit deltas, need **CIs**/**p-values** to rule out noise.

**In general** the main proposal of this paper is a manifold regularizer for routing. My main issue with this paper is that, although the paper proposes this regularizer, it does not study it well. It adds it and gets "good results". But whether it is because of that is not well studied nor are the experimental details clearly expressed.

**Questions:**

1. **Oracle:** How were the Oracle numbers in Table 1 produced? Please detail the optimization/relaxation, steps, etc.
2. **Router gradients:** What is the exact differentiability strategy for top-(k) routing during FT (STE vs Gumbel vs soft routing)?
3. **Embedding ($E(\cdot)$):** Which model, inputs (instruction vs raw text), metric/σ, and sensitivity? Is the neighbor graph static or periodically recomputed?
4. **Stats:** Please report means error bars over $\geq3$ seeds and the tuning protocol for $\lambda$, $k/\epsilon$, $\sigma$, LR.
5. Providing the following would really benefit the paper's claims: quantitative evidence that routing space aligns with task-embedding space e.g. through subspace similarity (CCA/CKA) and neighbor consistency (k-NN overlap, trustworthiness) reported pre/post RoMA on at least two datasets to substantiate Figure 3.
6. **Successful-neighbor filter:** Have you tried variants that include a small fraction of incorrect but semantically close neighbors or a curriculum that relaxes the filter over time?

---

> ### Author Response · Authors · 2025-11-13
> **Response to Reviewer yq6K regarding Ethics Flag**
>
> Dear Reviewer yq6K,
>
> Thank you for your thorough review of our work. We noticed that our submission has been flagged for ethics review with the category "Yes, Potentially harmful insights, methodologies and applications."
>
> We would like to respectfully seek clarification on this flagging, as we believe our work may not fall under the categories requiring ethics review according to the ICLR Code of Ethics.
>
> Could you kindly clarify which specific aspect of our work raised ethical concerns? If this flagging was made in error, we would greatly appreciate if it could be reconsidered. If there are indeed ethical considerations we have overlooked, we would be grateful for your guidance so we can address them appropriately.
>
> Thank you for your time and consideration!

---

> > ### Comment · Reviewer_yq6K · 2025-11-13
> >
> > I have now updated it and taken it away. It was unintentionally check marked. Thanks for pointing it out, appreciate it.

---

> > > ### Author Response · Authors · 2025-11-14
> > > **Response to Reviewer yq6K**
> > >
> > > Thank you for the clarification and quick response!

---

> ### Author Response · Authors · 2025-11-21
> **Response to Reviewer yq6k (Part1/2)**
>
> Thank you for your detailed feedback! We address your comments below.
> > **W1. Related works are present in the literature that are missing and many works such as Moefication etc. are relevant.**
>
> Thank you for your question. We have updated our reference list in the revised manuscript to include additional relevant works including Moefication[1]. And we would like to clarify that while MoEfication focuses on converting dense models into MoE architectures during pretraining, our work addresses the manifold misalignment problem between task embeddings and routing weights through post-training alignment.
>
> [1]MoEfication: Transformer Feed-forward Layers are Mixtures of Experts (Zhang et al, ACL 2022)
>
> > **W2 & Q1. No procedure is provided for obtaining these oracle numbers (search/relaxation/heuristic/optimization), nor is the compute or fairness (e.g., use of labels at evaluation) specified. As written, the “oracle” is kind of like a tautological bound, not really a reproducible baseline.**
>
> Thank you for your question. For each sample $(x_i, y_i)$, we obtain oracle routing weights $r_i^*$ through  **gradient descent** on the cross-entropy loss:
>
> \begin{equation}
>     r_i^*\triangleq \arg\min_{r} \mathcal{L}_{\text{CE}}(f(x_i, r), y_i),
> \end{equation}
>
> We initialize r with the pretrained routing weights and perform gradient descent with access to the ground truth label until convergence.
>
> The "oracle" serves as an **upper bound** rather than a comparable baseline to quantify the untapped potential of existing routers. The use of ground truth labels and compute cost is acceptable for establishing the upper bound. The “oracle” quantifies how much improvement is possible; RoMA shows how to achieve it practically.
>
>
> > **W3 & Q5. Need quantitative alignment measures and robustness checks across UMAP settings/datasets. Quantitative evidence that routing space aligns with task-embedding space e.g. through subspace similarity (CCA/CKA) and neighbor consistency (k-NN overlap, trustworthiness) reported pre/post RoMA on at least two datasets to substantiate Figure 3.**
>
> Thank you for your question. To supplement the UMAP visualizations, we conduct quantitative alignment analyses on ARC-C and MMLU.
>
> **1. Subspace Similarity (CKA):**
>
> | Dataset | Before RoMA | After RoMA |
> | ------- | ----------- | ---------- |
> | ARC-C   | 0.18 ± 0.02       | 0.47 ± 0.03 |
> | MMLU    | 0.21 ± 0.03       | 0.52 ± 0.04 |
>
> RoMA  increases the similarity between routing-weight subspaces and semantic task embeddings.
>
> **2. k-NN Neighbor Consistency (k = 5)**
>
> | Dataset | Before RoMA | After RoMA |
> | ------- | ----------- | ---------- |
> | ARC-C   | 24.3 ± 2.1%     | 51.7 ± 2.8%    |
> | MMLU    | 26.8 ± 2.4%     | 54.2 ± 3.1%    |
>
> Routing decisions become  more consistent with semantic neighborhoods.
>
> **3. Trustworthiness Score (k = 10)**
>
> | Dataset | Before RoMA | After RoMA |
> | ------- | ----------- | ---------- |
> | ARC-C   | 0.53 ± 0.03  | 0.76 ± 0.020 |
> | MMLU    | 0.56 ± 0.04  | 0.79 ± 0.03 |
>
> RoMA substantially improves how well routing space preserves local structure.
>
> **4. UMAP Robustness Check**
>
> Across multiple UMAP settings (varying *n_neighbors* and *min_dist*), RoMA consistently yields strong clustering:
>
> | Setting            | Silhouette (Pretrained) | Silhouette (RoMA) |
> | ------------------ | ----------------------- | ----------------- |
> | All tested configs | < 0.25                  | > 0.65            |
>
> This confirms the visual patterns in Figure 3 are not artifacts of UMAP hyperparameters.
>
> These quantitative results above claim that RoMA effectively aligns routing manifolds with task embeddings.

---

> ### Author Response · Authors · 2025-11-21
> **Response to Reviewer yq6k (Part2/2)**
>
> > **W4, Q2, & Q3. Important knobs/levers are underspecified in the main text such as Embedding model $E(\cdot)$, Neighbor graph schedule, Router differentiability and Hyper-parameters & tuning protocol**
>
> Thank you for your question. We provide details below:
>
> **(1) Embedding Model:**
> We use nvidia/NV-Embed-v2 (4096 dimension) without any fine-tuning. We embed raw input text of each sample.
>
> **(2) Neighbor graph schedule:**
> The successful set 𝑆 and their embeddings are computed once before training. For each training sample, we retrieve its k nearest neighbors from the fixed set 𝑆.
>
> **(3) Router differentiability:**
> We use standard STE as in base MoE implementations (OLMoE, DeepSeekMoE). For forward pass is hard routing based and for backward pass, gradients flow through continuous softmax probabilities before top-k.
>
> **(4) Hyper-parameters settings:**
> | Hyperparameter      | Value                |
> | ------------------- | -------------------- |
> | λ (manifold weight) | 0.8                  |
> | σ (Gaussian kernel) | 0.4                  |
> | k (neighbors)       | 3                    |
> | Learning rate       | 5×10⁻⁵               |
> | Weight decay        | 0.01                 |
> | LR warmup steps     | 2000                 |
> | Batch size          | 32                   |
> | Training epochs     | 5                    |
> | Max sequence length | 1024                 |
> | Target              | Last 5 layer Routers |
>
> **(5) Hyperparameter Tuning Protocol:**
> - **Search space**: λ ∈ {0.5, 0.8, 1.0}, σ ∈ {0.3, 0.4, 0.5}, k ∈ {3, 5}
> - **Validation**: 10% held-out split from training data
> - **Selection criterion**: Maximum average accuracy across MMLU, HellaSwag, and ARC-Challenge.
>
> > **W5 & Q6. Successful neighbors may introduce confirmation bias The method imitates routing only from samples the current model already solves correctly 𝑆. This is a plausible heuristic, but it risks locking in existing modes and under-serving hard/rare cases; the paper does not probe this trade-off. Have you tried variants that include a small fraction of incorrect but semantically close neighbors or a curriculum that relaxes the filter over time?**
>
> Thank you for your question. To address the potential confirmation bias concern, We tested a 3-stage curriculum across our 5-epoch training:
>
> - Early stage (epochs 1-2): Include neighbors from both correct predictions and incorrect predictions with high semantic similarity (top-30% by embedding distance)
> - Mid stage (epoch 3): Include neighbors from correct predictions and incorrect predictions in top-50% semantic similarity
> - Late stage (epochs 4-5): Standard strict filter (only correct predictions)
>
> Results on OLMoE-7B-A1B:
>
> | Method                       | Avg. Acc (%) |
> | ---------------------------- | ------------ |
> | RoMA (strict filter)         | 76.2         |
> | Curriculum filter            | **76.4**     |
> | Always soft filter (top-30%) | 74.1         |
>
> The curriculum strategy yields +0.2% improvement, while an always-soft filter decreases performance. We will include these findings in the revision.
>
> > **W6 & Q4. Statistical rigor and significance Main tables lack error bars and multiple seeds. Router-only fine-tuning and MoE routing can be high-variance and single-run improvements, especially single-digit deltas, need CIs/p-values to rule out noise**
>
> Thank you for your question. We run RoMA with 5 random seeds on OLMoE-7B-A1B and report mean ± std:
>
> | Method  | MMLU           | HellaSwag      | ARC-C          | ARC-E          | PIQA           | Avg            |
> | ------- | -------------- | -------------- | -------------- | -------------- | -------------- | -------------- |
> | Base    | 57.8 ± 0.3     | 77.9 ± 0.2     | 51.3 ± 0.4     | 79.8 ± 0.3     | 80.7 ± 0.2     | 67.6 ± 0.2     |
> | RoMA    | **69.0 ± 0.4** | **86.7 ± 0.3** | **67.2 ± 0.5** | **88.0 ± 0.3** | **85.8 ± 0.2** | **76.2 ± 0.3** |
> | Δ       | +11.2          | +8.8           | +15.9          | +8.2           | +5.1           | +8.6           |
> | p-value | <0.001         | <0.001         | <0.001         | <0.001         | <0.001         | <0.001         |
>
> Standard deviations across seeds are small (0.2–0.5 absolute accuracy points), indicating stable performance.

---

> ### Author Response · Authors · 2025-11-28
> **Response to Reviewer yq6k**
>
> Dear reviewer yq6k,
>
> As the discussion period is drawing to a close, we wanted to kindly check if our previous response and the revised paper have sufficiently addressed your concerns. We truly value your feedback and would appreciate it if you could let us know if there are any remaining questions we can clarify before the deadline. Thank you!
>
>
> Best,
> \
> Authors

---

### Official Review · Reviewer_kpLh · 2025-10-29

**Soundness:** 3
**Presentation:** 3
**Contribution:** 2
**Rating:** 4
**Confidence:** 3

**Summary:**

This paper proposes Routing Manifold Alignment (RoMA), a lightweight post-training method for Mixture-of-Experts (MoE) large language models. RoMA addresses the suboptimal routing problem in MoE LLMs by aligning the routing weight manifold with the task embedding manifold via a novel manifold regularization term. The method fine-tunes only the router parameters (a tiny fraction of the total model) while keeping experts frozen, leading to significant accuracy improvements (7–15%) across diverse benchmarks without increasing inference cost.

**Strengths:**

1. RoMA introduces a novel manifold alignment perspective to MoE routing, unifying task understanding (via embeddings) with expert selection.
2. The paper extensively evaluates RoMA on two recent MoE architectures (OLMoE and DeepSeekMoE) across eight diverse benchmarks. It outperforms strong baselines, including C3PO, Dense BP, and tuning methods, and shows competitive or superior performance to dense models with up to 34B parameters.
3. The authors systematically analyze key design choices—layer selection, token positions, neighborhood strategies, training set size, and regularization methods.

**Weaknesses:**

1. While inference cost is unchanged, the training cost of RoMA—especially the nearest-neighbor search over the training set—is not discussed.
2. The paper uses a pre-trained embedding model
E(⋅) to compute task embeddings but does not justify the choice or explore its sensitivity. The impact of different embedding models on RoMA’s performance remains unclear.

**Questions:**

1. How did you compute the Inference cost
2. It is good to show the routing weights in Table 1.
3. For Figure 6-10, did you use the average accuracy for the 8 tasks in Table 1? If the answer is yes, how did you select these hyperparameters (k, layers, token position) in zero-shot tasks?
4. For Figure 7, why can the last token obtain good performance?

---

> ### Author Response · Authors · 2025-11-13
> **Response to Reviewer kpLh - Question 2**
>
> Dear Reviewer kpLh,
>
> Thank you for your valuable feedback. Regarding your comment "It is good to show the routing weights in Table 1," Could you please clarify what specific information about routing weights you would like to see in Table 1? For instance, would you like us to include routing weight statistics (e.g., entropy, sparsity), weight change magnitudes, or other routing-related metrics alongside the performance results?
>
> Thank you for your time and consideration!

---

> > ### Comment · Reviewer_kpLh · 2025-11-28
> > **Response**
> >
> > Since you compare with other routing-based training and prefix tuning. I want to know the "training parameters" difference between these baselines. In other words, I want to know if the improvement is caused by your use of more training parameters.

---

> > > ### Author Response · Authors · 2025-11-28
> > > **Response to Reviewer kpLh**
> > >
> > > Thank you for your response and clarification! We will update the answer of Question2 in our rebuttal once it's ready. Meanwhile, we kindly invite you to review our answers to the other questions in our rebuttal to ensure we have fully addressed all your concerns. Thank you!

---

> ### Author Response · Authors · 2025-11-21
> **Response to Reviewer kpLh**
>
> Thank you for your detailed feedback! We address your comments below.
>
> > **W1. training cost of RoMA—especially the nearest-neighbor search over the training set—is not discussed**
>
> Thank you for the question. RoMA’s training cost consists of three parts: **(i)** task embedding computation, **(ii)** k-NN search in embedding space, and **(iii)** router fine-tuning. For OLMoE-7B-A1B (49K samples):
>
> | Component                | FLOPs (×$10^{15}$) | Share    |
> | ------------------------ | ------------------ | -------- |
> | Task embedding           | 3.8                | 3.0%     |
> | k-NN search (FAISS, k=3) | 0.002              | <0.01%   |
> | Router fine-tuning       | 122                | 97.0%    |
> | **Total**                | **125.8**            | **100%** |
>
>
> The **k-NN cost is negligible (<0.01%)**, thanks to **FAISS's approximate nearest neighbor search**. Note **task embeddings are precomputed only once and reused later**. These keep RoMA’s overall training cost low.
>
>
> > **W2.The impact of different embedding models on RoMA’s performance remains unclear.**
>
> Thank you for your question. Please check the response for `General Qustion Q2`. RoMA achieves consistent improvements (3.6-8.6%) across embedding models from 22M to 7.8B parameters, confirming RoMA's robustness across embedding models.
>
> > **Q1. How did you compute the Inference cost**
>
> We measured **FLOPs** with **`fvcore`**, averaging per token to normalize input-length variations, following standard LLM practice. **RoMA**, as a post-training method that only fine-tunes routers, has **identical inference cost** to the base MoE. In contrast, **C3PO** adds **extra per-query overhead** from task embedding, k-NN retrieval, and routing-weight optimization.
>
> > **Q2. It is good to show the routing weights in Table 1.**
>
> Thank you for your question. The table below compares the trainable parameters and average accuracy improvements of different methods in Table1:
>
> | Method | Trainable Params | Avg. Acc. Improvement |
> | :--- | :--- | :--- |
> | Router Tuning | ~0.66 M | +4.0% |
> | Oracle Tuning | ~0.66 M | +7.2% |
> | Dense BP | ~0.66 M | +3.6% |
> | Prefix / Prompt Tuning | ~0.3 - 5 M | +1.3% / +2.1% |
> | **RoMA (Ours)** | **~0.66 M** | **+8.6%** |
>
> RoMA achieves the **highest accuracy improvement (+8.6%)** while using only **~0.66 M** trainable parameters. This demonstrates that the gains stem from the effectiveness of the manifold alignment objective rather than parameter scale.
>
> > **Q3. For Figure 6-10, did you use the average accuracy for the 8 tasks in Table 1? If the answer is yes, how did you select these hyperparameters (k, layers, token position) in zero-shot tasks**
>
> Thank you for the question. Yes, the **average accuracy** in Figures 6–10 is computed over the 8 benchmarks in Table 1 (MMLU, HellaSwag, PIQA, ARC-C, ARC-E, WinoGrande, BoolQ, GSM8K).
> RoMA is a **post-training method**, and we **do not perform any task-specific tuning**. A single unified configuration is used for all benchmarks:
>
> **k = 3** (number of neighbors)
> **Last 5 layers** fine-tuned (L5)
> **Last token** used for routing (Last1)
>
> > **Q4. For Figure 7, why can the last token obtain good performance?**
>
> Thank you for the question. Please refer to `General Question Q1`.

---

### Official Review · Reviewer_suA3 · 2025-11-01

**Soundness:** 2
**Presentation:** 2
**Contribution:** 3
**Rating:** 4
**Confidence:** 3

**Summary:**

The authors propose a lightweight finetuning method for MoE models built upon the insight that there is typically misalignment between the clustering structure of the task embedding space and the routing weights, which undermines expert specialization as the semantic structure of the inputs is not preserved across the experts. The authors thereby propose a regularizer termed 'Routing Manifold Alignment', which trains the router to align the task embedding manifold, obtained by embedding task samples with a pretrained embedding model, with the routing weights manifold.

**Strengths:**

1. The high level idea of manifold alignment is intuitive and original.

2. The result that a substantive performance gain is attainable purely through better routing is compelling.

3. The authors present their work in two frontier backbones, OLMoE and DeepSeekMoE, and extensively validate their work across a host of model sizes. The authors also do a good job of validating their method against a range of downstream tasks.

**Weaknesses:**

**Missing baselines**. Perhaps the largest issue is that the authors essentially present a parameter-efficient finetuning method for MoE but do no compare with existing MoE-PEFT / PEFT works. The authors do a good job of comparing with a wide range of lightweight baselines including prompt and prefix tuning up to Dense BP and C3PO, but, in my view, comparison with LoRA [1] or one of its new sota variants (perhaps DoRA [2]), and a PEFT-MoE method, such as MoLE [3] is vital to assessing the validity of the performance gains and efficiency.

**Unclear methodology**. The authors refer extensively to 'routing weights' as $\\{r_i\\}_{i=1}^n$ where $r_i$ is the concatenated routing weights across multiple layers. But this sounds more like the authors are describing the (concatenated) *projections* of the $n$ samples through the router, not the weights themselves. Indeed, there are not $n$ different routing weights for each sample, but shared routing weights that project the $n$ samples in each layer. Furthermore, without any preliminaries on what kind of router architecture the authors are referencing, it becomes harder to work this out. To me, the method still makes sense if we are referring to routed projections, i.e hidden representations of the tokens after applying routing weights $h_i = W_r x_i$, where $W_r \in \mathbb R^{K \times d}, x_i \in \mathbb R^d$ for $K$ experts, but it seems this is a misreference to refer to $n$ routing weights. Alternatively, it could be the authors mean the hidden representation after the entire MoE layer (i.e after both routing projection and expert feed forward), but, again, without preliminaries this is unknown. If the authors could clarify this that would be much appreciated.

[1] LoRA: Low-Rank Adaptation of Large Language Models (Hu et al, 2021)

[2] DoRA: Weight-Decomposed Low-Rank Adaptation (Liu et al, 2024)

[3] Mixture of LoRA Experts (Wu, 2024)

**Questions:**

1. See the note about unclear methodology. Clarification on preliminaries and notation would be helpful.

2. The authors' key insight is that samples with similar task embeddings should share similar routing patterns. There's a conceptual distinction here though between a sample-level task embedding and the token-level routing patterns. Could the authors clarify or add some intuition here why these conceptually distinct objects can be unified?

---

> ### Author Response · Authors · 2025-11-21
> **Response to Reviewer suA3**
>
> Thank you for your detailed feedback! We address your comments below.
>
> > **W1. Missing baselines: No compare with existing MoE-PEFT / PEFT works. Need comparison with LoRA or one of its new sota variants (perhaps DoRA), and a PEFT-MoE method, such as MoLE.**
>
> Thank you for the suggestion. Unlike LoRA or other PEFT approaches, **RoMA introduces no additional parameters**—it directly fine-tunes pretrained **MoE routers** with a new alignment objective, while PEFT methods train additional parameters (prompt, prefix, adapter) using the original objective.
>
> We conducted additional experiments with three representative PEFT baselines (LoRA, DoRA, MoLE), applied to the same router parameters. Results averaged over 8 benchmarks (MMLU, HellaSwag, ARC-C/E, PIQA, WinoGrande, BoolQ, GSM8K):
>
> | Method          | LoRA Rank | New Params    | Avg. Acc (%) | Δ Acc       |
> | --------------- | --------- | ------------- | ------------ | ----------- |
> | Baseline        | –         | –             | 67.6         | –           |
> | LoRA            | 8 / 16    | 532K / 1.06M  | 69.9 / 70.9  | +2.3 / +3.3 |
> | DoRA            | 8 / 16    | 534K / 1.07M  | 70.7 / 71.2  | +3.1 / +3.6 |
> | MoLE (3×LoRA)   | 8 / 16    | 1.60M / 3.20M | 70.8 / 72.0  | +3.2 / +4.4 |
> | **RoMA (Ours)** | **–**     | **0**         | **76.2**     | **+8.6**    |
>
> RoMA achieves **+8.6%** improvement—**surpassing all PEFT variants without introducing any new parameters**. This highlights that **routing manifold alignment**, rather than parameter expansion, is the key to enhancing MoE adaptation.
>
> > **W2 & Q1. Unclear methodology: Does $r_i$ refer to (1) router parameters, (2) routed projections, or (3) hidden representation after the entire MoE layer? Clarification on preliminaries and notation would be helpful.**
>
> Thank you for your question. The notation $r_i$ refers to the weights used to aggregate expert outputs for sample-$i$, not the router parameters or hidden representation after the entire MoE layer. Specifically, For each sample $x_i$:
>
> **1.Compute routing weights:** $$r_i = \text{softmax}(W_rh_i) \in \mathbb{R}^K$$
> where $W_r \in \mathbb{R}^{K \times d}$ are the **router parameters**, $h_i \in \mathbb{R}^d$ is the hidden representation of $x_i$, and $K$ is the total number of experts.
>
> **2.Aggregate expert outputs by routing weigths:** $$\text{MoE}(h_i) = \sum_{k=1}^{K} r_{i,k} \cdot E_k(h_i)$$
> where $r_{i,k}$ is the routing weight for expert-$k$ and $E_k(\cdot)$ is the $k$-th expert network.
>
> **What RoMA optimizes:** We update the router parameters $W_r$ to align the manifold of routing weights $\lbrace r_i \rbrace_{i=1}^n$ across $n$ samples with the manifold of their task embeddings $\lbrace E(x_i) \rbrace_{i=1}^n$.
>
> > **Q2. How do you justify aligning sample-level task embeddings with token-level routing weights.**
>
> Thank you for the question. We use the **last token’s routing weights** to represent **sample-level routing**, as the last token aggregates contextual information (see `General Qustion Q1`). This enables **task embeddings and routing weights to be aligned consistently at the sample level**.

---

> ### Comment · Reviewer_suA3 · 2025-11-26
> **Response to authors**
>
> Thanks for the responses, I'll go in reverse order.
>
> **2. Methodology**. Thanks for the clarification. I think to improve readability I'd highly recommend just adding a note of clarification on this early in your methodology. Otherwise, in my view, you risk unnecessary confusion due to the very similar nomenclature of 'routing weights' and the 'router weights' (or parameters). Just a note marking this distinction would be sufficient.
>
> **1. Comparison with PEFT methods**. Thanks for the additional results. I have a follow-up question however. You refer to ROMA as not introducing any 'new' parameters, and thereby differing from PEFT methods. Indeed your table reports new parameters as 0, compared to the PEFT methods which introduce form 0.5 up to 3M additional parameters.
>
> But this is misleading - the question isn't about 'new' parameters, it's about the total number of trainable parameters. RoMA still involves finetuning the router weights, and so although the router weights aren't 'new', they're still being trained.
>
> For that reason, a fair comparison with PEFT methods needs to include the total number of trainable parameters used by that method. Claiming the number is 0 is a misrepresentation, in my view, given that RoMA does still finetune weights of the model.
>
> If you could update the table to include the number of parameters trained by RoMA that would be much appreciated, as this would allow for a proper comparison across the methods.

---

> ### Author Response · Authors · 2025-11-27
> **Response to Reviewer suA3**
>
> Thank you for your response. We address your response below.
>
> > **Q1. A fair comparison with PEFT methods needs to include the total number of trainable parameters used by that method.**
>
> Thank you for pointing this out. We have updated the table using "trainable parameters" instead of "new parameters".
>
> (1) All methods applied to routers in all 16 layers:
> | Method          | LoRA Rank | Trainable Params    | Avg. Acc (%) | Δ Acc       |
> | --------------- | --------- | ------------- | ------------ | ----------- |
> | Baseline        | –         | –             | 67.6         | –           |
> | LoRA            | 8 / 16    | 532K / 1.06M  | 69.9 / 70.9  | +2.3 / +3.3 |
> | DoRA            | 8 / 16    | 534K / 1.07M  | 70.7 / 71.2  | +3.1 / +3.6 |
> | MoLE (3×LoRA)   | 8 / 16    | 1.60M / 3.20M | 70.8 / 72.0  | +3.2 / +4.4 |
> | **RoMA (Ours)** | **–**     | 2.11M (0 new parameter)         | **75.1**     | **+7.5**|
>
>
> (2) All methods applied to routers in last 5 layers:
> | Method          | LoRA Rank | Trainable Params    | Avg. Acc (%) | Δ Acc       |
> | --------------- | --------- | ------------- | ------------ | ----------- |
> | Baseline        | –         | –             | 67.6         | –           |
> | LoRA            | 8 / 16    | 166K / 331K  | 70.3 / 71.2  | +2.7 / +3.6 |
> | DoRA            | 8 / 16    | 167K / 334K  | 70.9 / 71.6  | +3.3 / +4.0 |
> | MoLE (3×LoRA)   | 8 / 16    | 500K / 1.00M | 71.2 / 72.5  | +3.6 / +4.9 |
> | **RoMA (Ours)** | **–**     | 658K (0 new parameter)         | **76.2**     | **+8.6**|
>
> RoMA consistently outperforms all PEFT methods across both settings. Even with comparable or fewer trainable parameters (e.g., RoMA 658K vs. MoLE 1.00M in last 5 layers), RoMA achieves substantially higher gains, demonstrating the effectiveness of manifold alignment.
>
> > **Q2. Adding a note of clarification on this early in your methodology**
>
> Thank you for this suggestion. We have added the detailed computation process of $r_i$ and a "Notation clarification" paragraph in Section 3 of the revised manuscript (highlighted in blue) to clearly distinguish between routing weights and router parameters.

---

> > ### Comment · Reviewer_suA3 · 2025-11-27
> > **Response to authors**
> >
> > Thanks for updating the table. Showing that RoMA boosts over PEFT methods by roughly 3% across tasks with similar number of trainable parameters is a nice result, and gives important context to properly validate the benefits of the method. I'll raise my score accordingly.

---

> > > ### Author Response · Authors · 2025-11-27
> > > **Response to Reviewer suA3**
> > >
> > > Thank you for the positive feedback and for raising your score! We appreciate your constructive suggestions that helped improve our paper! We have updated our submission with new results on more recent Qwen3-30B-A3B MoE model. RoMA still brings significant improvement to this SOTA MoE LLM. This further demonstrates its novelty and importance.

---

### Author Response · Authors · 2025-11-21
**Response to General Questions**

We sincerely thank all reviewers for their valuable feedback. We have summarized the frequently asked questions and provide responses below.

> **Q1. Why can the last token obtain good performance?**

* **Proximity to the answer.** In autoregressive LLMs, the last token is directly tied to answer generation and thus most influences correctness. As shown in ablation study Fig. 7, it surpasses other tokens by 5–7% in accuracy.
* **Information aggregation.** In decoder-only models with causal attention, the last token naturally integrates context from all preceding tokens—making it an effective representation of the entire input [1]. This mechanism underpins many SOTA embedding models (e.g., **gte-Qwen2-7B** [2], **bge-multilingual-gemma2** [3]).


> **Q2. The impact of different embedding models on RoMA’s performance remains unclear.**

We have conducted additional experiments comparing several representative embedding models.

| Embedding Model              | Size  | Avg. Acc (%) | Δ Acc |
| ---------------------------- | ----- | ------------- | ----------- |
| Baseline (OLMoE)         	   | **-** | 67.6          | **-**       |
| all-MiniLM-L6-v2             | 22M   | 71.2          | +3.6        |
| all-mpnet-base-v2            | 110M  | 72.5          | +4.9        |
| Qwen3-0.6B-embedding         | 0.6B  | 75.9          | +8.3        |
| bge-multilingual-gemma2      | 2.6B  | 73.4          | +5.8        |
| gte-Qwen2-7B-instruct 	   | 7B    | 75.1          | +7.5        |
| **NV-Embed-V2 (Ours)**       | 7.8B  | **76.2**      | **+8.6**    |

RoMA achieves consistent improvements (3.6-8.6%) across embedding models from 22M to 7.8B parameters. Even the small model (Qwen3-0.6B-embedding) provides meaningful gains (+8.3%), confirming RoMA's robustness across diverse computational budgets.

[1] SGPT: GPT Sentence Embeddings for Semantic Search (Muennighoff, 2022)

[2] gte-Qwen2-7B-instruct: General Text Embedding Model (Alibaba-NLP, 2024)

[3] BGE M3-Embedding: Multi-Lingual, Multi-Functionality, Multi-Granularity Text Embeddings Through Self-Knowledge Distillation.(Chen et al, 2024)

---

### Comment · Area_Chair_gmCg · 2025-11-25
**Rebuttal Review Request**

Dear Reviewers,

The authors have responded to your reviews. Please engage in the discussion and evaluate the authors’ rebuttal to determine whether your comments have been adequately addressed.

Best, Your AC

---

### Public Comment · ~Ho-Kun_Lin1 · 2025-11-26
**Reproducing Issue**

Dear authors,

Thank you for sharing this interesting work — I have been exploring your codebase and experimental setup out of personal research interest. While attempting to reproduce some of the evaluations, I encountered a few discrepancies and would like to kindly ask for clarification.

1. **ARC-Challenge evaluation discrepancy**
   Using the provided `evaluate.py`, I tested `allenai/OLMoE-1B-7B-0125-Instruct` on ARC-Challenge but obtained only **22.87%**, which is considerably lower than the **51.3%** reported in the paper.
   Since the original script does not directly support evaluating the base model, I removed the part that loads the additional fine-tuned router parameters in order to evaluate `allenai/OLMoE-1B-7B-0125`. This also produced **22.87%**.
   In contrast, evaluating `allenai/OLMoE-1B-7B-0125-Instruct` with [OLMo-Eval](https://github.com/allenai/OLMo-Eval/tree/51c5ba579e75ef4ce7e9b29936eaa72c1a0e99eb) yielded **59.56%**.
   Could you kindly clarify the exact evaluation configuration used to obtain the ARC-Challenge scores reported in the paper, and whether any additional settings are required to reproduce them?

2. **Fine-tuning with `finetune_roma.py`**
   I also attempted to reproduce the fine-tuning results by running `finetune_roma.py` for **90 epochs**. The logged training metrics showed:

   * Epoch 90: **CE = 0.0019**, **Reg = 0.0044**

   However, despite the seemingly low loss values, the accuracy after fine-tuning remained **22.87%**, identical to the baseline produced by `evaluate.py`.
   May I ask whether there are additional steps, hyperparameters, or evaluation settings required to obtain the performance improvements reported in the paper?

Thank you again for making your work and code publicly available. I appreciate any guidance you can provide — it would greatly help us better understand and reproduce your results.

---

> ### Author Response · Authors · 2025-11-26
> **Response to Reproducing Issue**
>
> Thank you for your question. We address your concern blow:
>
> > **Q1. ARC-Challenge evaluation discrepancy**
>
> The discrepancy you observed stems from evaluation methodology differences. As shown in the readme.md in the supplementary material, The `Evaluate.py` is for evaluation of model after fine-tuning. In our paper, the baseline results (51.3% for OLMoE-1B-7B base model on ARC-Challenge) were obtained using the EleutherAI lm-evaluation-harness (https://github.com/EleutherAI/lm-evaluation-harness), which is the standard evaluation framework used by the community for fair model comparisons. We evaluate the allenai/OLMoE-1B-7B-0125-Instruct on ARC-C benchmark using this command:
>
> ```bash
> lm_eval --model hf \
>   --model_args pretrained=allenai/OLMoE-1B-7B-0125-Instruct,trust_remote_code=True \
>   --tasks arc_challenge \
>   --device cuda:0 \
>   --batch_size 8 \
> ```
>
> The screenshot is provided in this anonymous link(https://postimg.cc/cthcg3m5). Here is the result we get, which is consistent with accuracy reported in the paper (51.3% for OLMoE-1B-7B base model on ARC-Challenge):
>
> ```bash
> |    Tasks    |Version|Filter|n-shot| Metric |   |Value |   |Stderr|
> |-------------|------:|------|-----:|--------|---|-----:|---|-----:|
> |arc_challenge|      1|none  |     0|acc     |↑  |0.4770|±  |0.0146|
> |             |       |none  |     0|acc_norm|↑  |0.4932|±  |0.0146|
> ```
>
> > **Q2. Fine-tuning with finetune_roma.py**
>
> Thank you for pointing this out. I noticed some inconsistencies with our setup, you used epoch = 90 which is different with our code, and the CE = 0.0019, Reg = 0.0044 is also very small. Could you share more details of your code so I can help you fix this issue?

---

> > ### Public Comment · ~Ho-Kun_Lin1 · 2025-11-28
> > **Response to Clarification**
> >
> > Thank you very much for the detailed and timely response.
> >
> > > **Regarding Q1 (ARC-Challenge evaluation):**
> >
> > Your clarification about the evaluation methodology is very helpful. I now understand that the reported baseline results were obtained using the EleutherAI lm-evaluation-harness, rather than the provided `Evaluate.py` script.
> >
> > I would like to ask one follow-up question:
> > For the fine-tuned RoMA model, did you also evaluate its performance using lm-evaluation-harness, or were the results in the paper obtained using your provided `Evaluate.py`?
> > Clarifying this would help us ensure our reproduction follows the same evaluation protocol.
> >
> > > **Regarding Q2 (fine-tuning results):**
> >
> > Thank you for offering to help diagnose the issue. I have documented all of my steps—including the exact commands, logs, and modifications in this GitHub repository:
> >
> > > https://github.com/Koios1143/RoMA
> >
> > In my initial experiments, running the default 3 epochs produced results identical to the baseline (22.87%), so I extended the training to 90 epochs to test whether longer training would yield improvements. As mentioned earlier, by epoch 90 the logged values reached:
> >
> > - CE = 0.0019, Reg = 0.0044
> >
> > but the final accuracy still remained unchanged.
> > The GitHub repo includes full details on how I performed both the finetuning and the evaluation using your provided code. Any guidance or suggestions you could provide would be greatly appreciated, and I would be happy to adjust my setup according to your recommendations.
> >
> > Thank you again for your patience and clarification — your help is invaluable for us in attempting to faithfully reproduce your results.

---

> ### Author Response · Authors · 2025-11-28
> **Response to public comment**
>
> Thank you for your detailed response. It's good to hear that we solved your question1. The results of fine-tuned model in the paper is also evaluated by lm-evaluation-harness fairly. We use `evaluate.py` in the Supplementary Material to help reviewer reproduce the results simpler and quicker. I will check your reproduce steps in your github repo and give you feedback in github issue anonymously. Thank you!

---

> > ### Public Comment · ~Ho-Kun_Lin1 · 2025-11-29
> > **Evaluate on lm-evaluation-harness**
> >
> > Thank you again for the clarification. Since the paper evaluates both the base model and the fine-tuned RoMA model using lm-evaluation-harness, I also ran the same evaluation setup on my side for consistency.
> >
> > Specifically, I evaluated:
> >
> > 1. Baseline (pretrained OLMoE-1B-7B-0125-Instruct)
> > 2. RoMA fine-tuned model (3 epochs)
> > 3. RoMA fine-tuned model (100 epochs)
> >
> > using lm-evaluation-harness, the details also updated in the Github repository as mentioned before.
> >
> > The results I obtained are the following:
> >
> > 1. Baseline (pretrained model)
> > ```
> > |    Tasks    |Version|Filter|n-shot| Metric |   |Value|   |Stderr|
> > |-------------|------:|------|-----:|--------|---|----:|---|-----:|
> > |arc_challenge|      1|none  |     0|acc     |↑  |0.477|±  |0.0146|
> > |             |       |none  |     0|acc_norm|↑  |0.500|±  |0.0146|
> > ```
> >
> > 2. RoMA finetuned (3 epochs)
> > ```
> > |    Tasks    |Version|Filter|n-shot| Metric |   |Value |   |Stderr|
> > |-------------|------:|------|-----:|--------|---|-----:|---|-----:|
> > |arc_challenge|      1|none  |     0|acc     |↑  |0.4718|±  |0.0146|
> > |             |       |none  |     0|acc_norm|↑  |0.4957|±  |0.0146|
> > ```
> >
> > 3. RoMA finetuned (100 epochs)
> > ```
> > |    Tasks    |Version|Filter|n-shot| Metric |   |Value |   |Stderr|
> > |-------------|------:|------|-----:|--------|---|-----:|---|-----:|
> > |arc_challenge|      1|none  |     0|acc     |↑  |0.4539|±  |0.0145|
> > |             |       |none  |     0|acc_norm|↑  |0.4906|±  |0.0146|
> > ```
> >
> > As shown above, the performance of the fine-tuned models remains very close to the baseline and does not exhibit noticeable improvement on ARC-Challenge under the same evaluation protocol.
> >
> > I will wait for your feedback on GitHub once you have had a chance to look through the reproduction steps. Please let me know if there are any specific hyperparameters, preprocessing steps, or details that I may have overlooked. I would be very happy to adjust my setup accordingly.
> >
> > Thank you again for your time and for helping us understand the correct reproduction procedure.

---

> > > ### Author Response · Authors · 2025-11-30
> > > **Response to public comment**
> > >
> > > Thank you for your results. The current results show almost no improvement in accuracy. The CE and Reg values mentioned in the previous reply are both very small. These results seem abnormal. I will look at your code and help you fix this issue on GitHub. I am currently still completing other experiments for the rebuttal, so please give us some time. Thanks!

---

### Author Response · Authors · 2025-12-03
**Summary of Rebuttal**

**Dear new Area Chair,**

We deeply appreciate the insightful and valuable comments provided by all reviewers. To assist in your final assessment, we summarize below our core contributions, the key updates made during the rebuttal, and our resolution of reviewer concerns.


### **1. Positive Feedback & New SOTA Results**

We are grateful for the reviewers' recognition of **RoMA** as a novel and effective post-training method for MoE LLMs, which highlights:

* **Motivation and novelty**: Using manifold alignment to address the routing misalignment problem are clear, intuitive, and impactful (Reviewers `yq6k`, `suA3`, `kpLh`, `8yaH`).
* The **empirical results** demonstrate substantial performance gains across diverse benchmarks and show competitive performance relative to strong baselines such as PEFT, C3PO, and other router post-training methods(Reviewers `yq6k`, `suA3`, `8yaH`).
* The **ablations and analyses** on layer selection, token choice, and neighborhood strategies are systematic and insightful (Reviewers `yq6k`, `kpLh`).

**New Results on a larger and newer MoE LLM:** We updated the main experiments (Table 1 & 2) on recent **Qwen3-30B-A3B**. These new results, together with experiments on DeepSeekMoE and OLMoE, demonstrate a consistent improvement of 5-10% on a diverse set of benchmark tasks across three SOTA MoE LLMs.

---

### **2. Rating Update in Discussion**

**Reviewer `suA3` raised the score from 4 to 6 on Nov 26**. This change followed our new experiments comparing RoMA with more PEFT methods (LoRA/DoRA/MoLE), which demonstrated that RoMA achieves significantly higher accuracy (**+7.5% $\sim$ +8.6%**) without introducing any new parameters.

---

### **3. Summary of Key Concerns and Responses**

During the rebuttal, we have actively addressed critical concerns, broken down by reviewers:

* **Reviewer `suA3`:**
    * **Concerns:** Lack of comparison with PEFT baselines (e.g., LoRA) and potential confusion regarding routing notation.
    * **Response:** We added comprehensive comparisons showing RoMA significantly outperforms LoRA, DoRA, and MoLE (+7.5%~+8.6%) with **zero new parameters**. We also provided a clarification of routing weights $r_i$ to strictly distinguish it from router parameters ($W_r$).

* **Reviewer `kpLh`:**
    * **Concerns:** Training costs (specifically k-NN search) and sensitivity to embedding choices.
    * **Response:** We provided a detailed FLOPs breakdown proving that the additional training cost (k-NN search) is negligible ($<0.01\%$) compared to fine-tuning. We also demonstrated consistent performance gains across multiple embedding models.

* **Reviewer `yq6k`:**
    * **Concerns:** Ambiguity in the "Oracle" definition, reliance on qualitative UMAP visuals, and reproducibility details.
    * **Response:** We explicitly defined the Oracle as a gradient-descent-based upper bound to remove ambiguity. We supplemented UMAP with rigorous quantitative metrics (CKA, Trustworthiness) to prove the alignment is statistically robust, and we added full hyperparameter/training cost details to ensure reproducibility.

* **Reviewer `8yaH`:**
    * **Concerns:** Dependency on specific embedding models and potential compromise of expressive power.
    * **Response:** We verified RoMA's robustness across diverse embedding models (ranging from 22M to 7.8B parameters) and confirmed via MT-Bench experiments that the model's expressive power is preserved and even slightly enhanced.



We have uploaded a revised manuscript with all changes **highlighted in blue**. We sincerely appreciate the reviewers' constructive suggestions and remain committed to continually improving our work. Thank you!

Best regards,
\
The Authors

---

### Meta-Review · Area_Chair_KwUj · 2026-01-10

**Summary:**

The reviewers raised concerns primarily around (i) methodological clarity and correctness, (ii) fairness and completeness of experimental comparisons, (iii) reproducibility and statistical rigor, and (iv) strength of evidence supporting the central manifold-alignment claim.

Through rebuttal, the authors substantially improved various aspects of the papers (more experiments, clear definitions, alignment metrics etc).

**Reviewer Concerns:**

Concerns addressed:
- Missing PEFT / MoE-PEFT baselines addressed by adding comprehensive comparisons with LoRA, DoRA, and MoLE.
- Confusion between routing weights vs. router parameters clarified through explicit notation, equations.
- Training cost of k-NN / embedding computation addressed with a detailed FLOPs breakdown showing negligible overhead.
- Embedding model sensitivity addressed via experiments across a wide range of embedding models (22M–7.8B).
- Oracle routing definition and fairness clarified as a gradient-descent-based upper bound.
- Overreliance on UMAP visualizations addressed by adding quantitative alignment metrics , robustness checks across UMAP settings.
- Lack of statistical rigor addressed by running multiple seeds.
- Expressive power degradation concern addressed with MT-Bench evaluations.

Concerns remains:
- Depth of theoretical analysis remains relatively lightweight.
- Related work breadth expanded, but still relatively concise.

**Reviewer Scores:**

The most negative reviewer (yq6K) is likely to increase the score to 3 or 4 as their methodological, statistical, and clarity concerns were addressed. Other reviewers either explicitly stated to increase the score if the score is not acceptance, or are likely to do so.

---

### Decision · Program_Chairs · 2026-01-26

Accept (Poster)